# Long-read sequencing of families reveals increased germline and postzygotic mutation rates in repetitive DNA

Michelle D. Noyes[1], Yang Sui [1], Youngjun Kwon [1], Nidhi Koundinya [1], Isaac Wong [1], Katherine M. Munson [1], Kendra Hoekzema[1], Jennifer Kordosky[1], Gage H. Garcia [1], Jordan Knuth [1], Alexandra P. Lewis[1] & Evan E. Eichler [1,2] ✉

Long-read sequencing improves sensitivity to discover variation in complex repetitive regions, assign parent-of-origin, and distinguish germline from postzygotic mutations. We applied Illumina, Oxford Nanopore Technologies, and PacBio sequencing to discover and validate de novo mutations in 73 children from 42 autism families (157 individuals). We assay 2.77 Gbp of the human genome, yielding on average 95 de novo mutations per transmission (87.5 single-nucleotide substitutions, 7.8 indels), with no significant difference in mutation rate or profile between probands and their unaffected siblings. Long reads increase de novo mutation discovery by 20-40% and double the mutations classified as early embryonic. The germline mutation rate is $1.30\times10^{-8}$ substitutions/base pair/generation; the postzygotic rate is $0.23\times10^{-8}$. These rates are significantly increased in repetitive DNA, where segmental duplication mutability is dependent on length and percent identity. Here, we show that enrichment in repeats occurs predominantly postzygotically, likely resulting from faulty DNA repair and interlocus gene conversion.

De novo mutations (DNMs) are variants unique to a child that are absent from its parents. They arise from mutational processes in the parental germline, such as double-stranded break repair or errors in DNA replication[1–5], although the relative contribution of replication errors has been recently called into question[6–8]. In addition, a small number of de novo variants arise in the rounds of cell division just after fertilization, early enough in development that they can still be detected in almost all tissues. These mutations have been called early embryonic[9,10] or gonosomal[1], but they are most commonly referred to as postzygotic mutations (PZMs) to distinguish them from germline DNMs arising during the process of parental gametogenesis[11–13]. The most common type of DNMs are single base-pair substitutions and small (<50 bp) insertions and deletions; previous short-read DNM discovery efforts in humans report an average mutation rate of at least 70 DNMs per individual per generation, over three quarters of which arise in the paternal germline[1,3,4]. PZMs are more challenging to identify

and validate, and the PZM rate has been typically estimated to be up to 10% of the germline rate[1,14–17].

Most studies of human de novo variation have depended on short-read sequencing (SRS) of parent–child trios or quads to both identify variants and assign them to parental haplotypes. Discovery is typically limited to approximately 84% of the genome where SRS data (150–200 bp) can be reliably mapped[18]. Regions with the largest and most identical blocks of repeats, such as segmental duplications (SDs), are typically excluded despite the fact that such regions are predicted to be more mutable[19,20]. Indeed, population genetic and DNM studies on single families have confirmed increased rates of DNM with distinct mutational signatures potentially consistent with the action of interlocus gene conversion, but to date there have been few studies from a diversity of families[11,21–23]. In addition to discovery, SRS limits parent-of-origin assignment typically performed using informative single-nucleotide polymorphisms (SNPs) or variants flanking the DNM that

[1]Department of Genome Sciences, University of Washington School of Medicine, Seattle, WA, USA. [2]Howard Hughes Medical Institute, University of Washington, Seattle, WA, USA. ✉e-mail: ee3@uw.edu

can be uniquely traced to one parent or the other. Because the density of SNPs (>1/1200 bp) typically exceeds the average length of SRS (150–200 bp), fewer than 20% of DNMs are routinely phased and assigned to parent-of-origin[3]. Long-read sequencing (LRS) can capture many of these informative SNPs on a single read and can therefore be used to phase more DNMs, which is also useful for distinguishing postzygotic from germline variants due to the availability of long-range haplotype information[24,25].

In this study, we set out to comprehensively identify human germline and postzygotic de novo variation across 73 children from 42 families (Supplementary Fig. 1a). For all 73 children, we leveraged long-read PacBio high-fidelity (HiFi) sequencing data derived from blood for variant discovery, and both long-read Oxford Nanopore Technologies (ONT) and short-read Illumina data for validation purposes (Supplementary Fig. 1b). These families are part of the Simons Simplex Collection (SSC, n = 40) and Study of Autism Genetics Exploration (SAGE, n = 2), and all but one have a proband affected with simplex autism, as the sequencing data for that family's proband were not available. They have been examined for de novo variation before using Illumina whole-exome and whole-genome sequencing data[19,20,26,27], but were selected for LRS because no genetic (monogenic or polygenic) cause had been previously identified[28]. Here, we quantify both germline and postzygotic DNMs in each child. Using LRS, we recover more PZMs than previous Illumina-based studies and assign the parent-of-origin to >97% of all DNMs. The LRS data allow us to evaluate the germline and PZM rates across different genomic regions, quantifying the enrichment of DNMs in hypermutable repetitive sequences, and to compare the mutational signatures.

## Results

### Germline and postzygotic single-nucleotide variants (SNVs)

We examined HiFi data derived from blood and cell lines for a total of 157 individuals from 42 families affected with simplex autism (31 quads and 11 trios, n = 73 children) for de novo variant discovery (Supplementary Fig. 1a, b). While primarily of European ancestry (n = 58), the cohort also includes eight children of Indigenous American, five of Asian (3 East Asian and 2 South Asian), and two of African ancestry. With T2T-CHM13v2.0 as a reference genome, we applied LRS variant callers to identify SNVs (see "Methods"), selecting those unique to a child for validation with two orthogonal sequencing technologies: ONT and Illumina. For a variant to pass validation, we require that it be observed in a child's blood-derived HiFi sequence reads in addition to either their ONT or Illumina reads, and that it be absent from both parents across all three types of read data. For samples with only cell-line-derived HiFi data (n = 9 and Supplementary Fig. 2), we required a higher read support threshold in blood-derived Illumina data. Additionally, we required that every de novo variant is unique to the sample in which it was called; any variant observed in the HiFi reads of unrelated individuals from these families was assumed to be either a segregating, under-called parental allele in the parent, or a recurrent sequencing error. Ultimately, 6030 de novo SNVs passed our validation criteria (see "Methods"), for an average of 82.6 SNVs/child (Supplementary Data 1).

We classified SNVs as germline or postzygotic in origin initially by haplotype assignment. A variant was considered to be germline if it was present in all the HiFi and ONT reads from its parental haplotype of origin, while a PZM was found only on a fraction of reads from a given haplotype. In other words, PZMs manifest as three or more haplotypes, while germline DNMs occur in regions with only two haplotypes where all reads from one parental haplotype share the DNM. Unlike SRS data, the nature of long reads make such distinctions readily possible[11]. In cases of conflict or ambiguity, the small number of remaining variants were classified by examining the allele balance (AB), or the fraction of reads with the de novo allele, across all three sequencing platforms (see "Methods", Fig. 1a,

Supplementary Fig. 3a, b). We classified 917 SNVs as postzygotic in origin (Fig. 1b), while the remaining 5113 SNVs likely arose in the parental germline (Fig. 1c). There was no significant relationship between sequencing depth in any platform and the total number of germline or postzygotic SNVs observed (Supplementary Fig. 4a). Further, we do not see a significant correlation between sample age and PZM count, indicating that the impact of postnatal somatic mutations on our callset is minimal (Supplementary Fig. 5).

Assessing AB for DNMs across all three sequencing technologies, we calculate the average germline mutation as AB = 0.48. In contrast, the average PZM AB is significantly reduced (AB = 0.22) (Fig. 1d). Although HiFi and Illumina data were derived from blood and ONT data were generated from cell lines to increase scaffolding potential across repeats, we find that AB is remarkably consistent across platforms: 85.7% of PZMs and 89.2% of DNMs do not differ between sequencing platforms (two-sided chi-squared test, p-value > 0.05). Notably, a small number (n = 11) of germline DNMs have AB = 1 across HiFi and at least one other sequencing platform, and all but two of these events fall in repetitive regions (4 in retrotransposable elements, 3 in SDs, and 2 in centromeres). Using the Ensembl Variant Effect Predictor[29] to annotate the most severe predicted consequence of each variant (Fig. 2a), we find potentially deleterious DNMs and PZMs. When comparing quads with both an ASD-affected proband and sibling, we see no significant enrichment of mutations in probands (Supplementary Fig. 6a), nor do we see a difference in the predicted number of deleterious DNMs or PZMs in probands (Supplementary Fig. 6b), although we note that these families were selected because probands did not harbor an obvious gene-disruptive mutation based on SRS. Within the coding regions of neurodevelopmental disorder (NDD)-related genes (Supplementary Data 2; see "Methods"), we identified one stop-gain and three missense DNMs in probands and six missense DNMs in unaffected siblings. There is no significant difference in the overall burden of NDD-related DNMs (n = 610) between probands and siblings (two-sided chi-square test, p-value = 0.8, OR = 0.97). The pathogenic stop-gain DNM, present in SYNGAP1, and two potential candidates in the promoters of DDX3X and POGZ were reported in Sui et al.[28]. Only two DNMs were located within 1 kbp of a de novo structural variant (SV) (Supplementary Data 3 and Supplementary Fig. 7a, b), and both lie in intronic regions.

We were able to assign 98.0% and 96.1% of germline and postzygotic SNVs, respectively, to parental haplotypes, a small but significant difference (two-sided two-sample test for equality of proportions, p-value = 0.000119) (Fig. 2b and Supplementary Data 4). As expected, we observe a significant enrichment of germline mutations on paternal haplotypes (two-sided Wilcoxon signed-rank test, p-value = $1.14 \times 10^{-13}$), with a 3.98:1 paternal:maternal ratio. Surprisingly, PZMs show a modest paternal bias (two-sided Wilcoxon signed-rank test, p-value = 0.030), with a 1.15:1 paternal:maternal ratio, although these proportions differ significantly, reflecting their different origins (two-sided Z-test, p-value < $2 \times 10^{-26}$). Across samples, we observe considerable variance in the paternal bias (an average of 51% of phased PZMs are on paternal haplotypes, with a standard deviation of 18%, Supplementary Fig. 8a). Even when applying more stringent filters, such as restricting to unique regions and requiring more read support, the paternal bias remains significant at 1.15:1 (Supplementary Fig. 8b). Germline mutations increase in number, by approximately 1.32 and 0.46 additional DNMs per year of paternal and maternal age, respectively (linear regression, p-value = $1.24 \times 10^{-10}$ and $3.32 \times 10^{-5}$) (Supplementary Fig. 9). Parental age has a more modest effect on postzygotic variation, with an additional 0.26 and 0.16 PZMs per year of paternal and maternal age, respectively (p-value = 0.011 and 0.14) (Fig. 2c). When we compare germline maternal and paternal SNVs, we see a significant paternal enrichment in transcription factor binding sites (maternal fraction 0.46, paternal fraction 0.51, two-sample test for equality of proportions p-value = 0.013), which is consistent with the

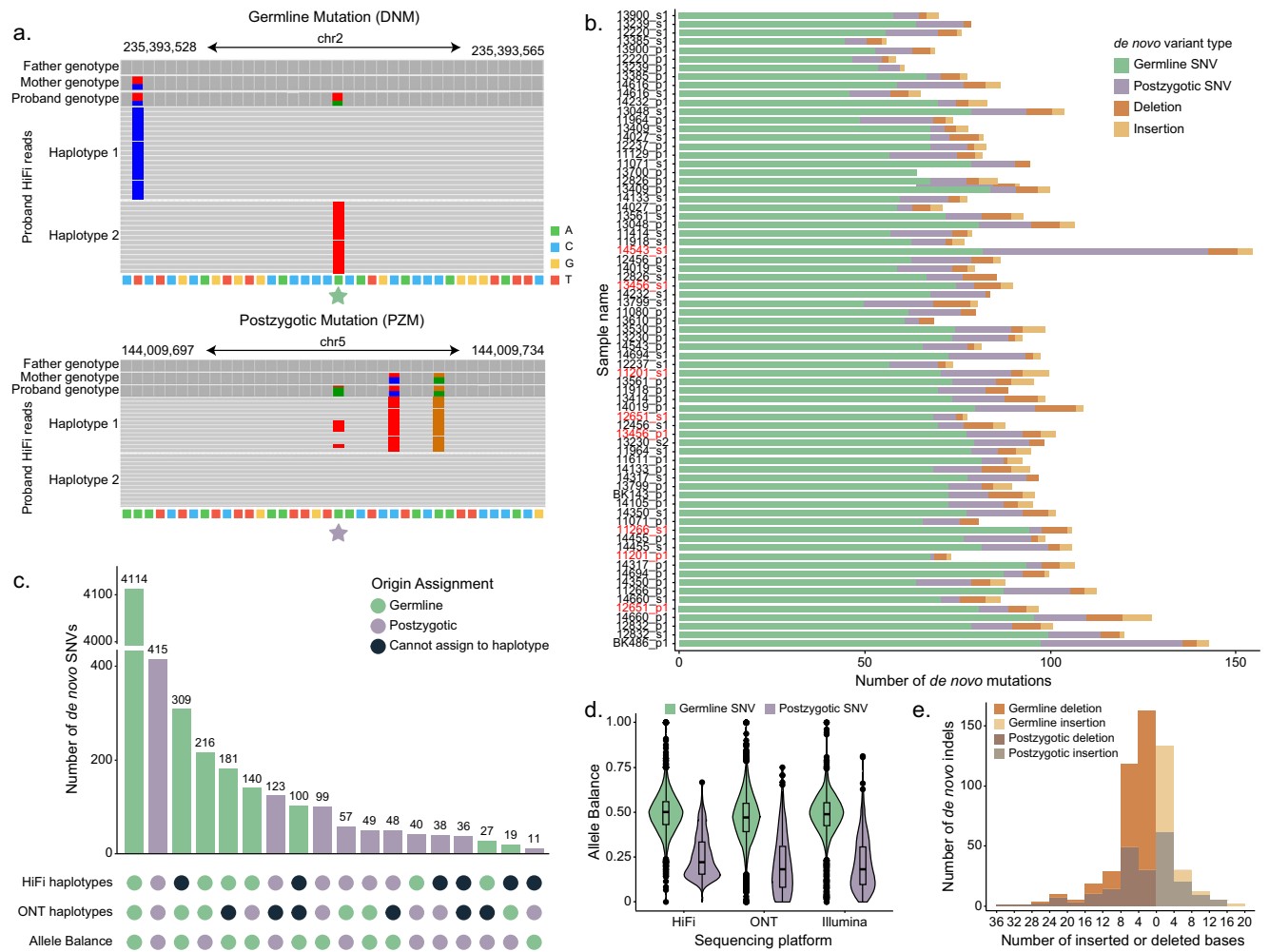

**Fig. 1 | Autosomal single-nucleotide variants (SNVs) and indels. a** Example IGV views of a germline and postzygotic mutation in HiFi read data. **b** The number of autosomal de novo germline and postzygotic SNVs, insertions and deletions <50 bp. Samples are ordered by increasing paternal age at birth (with the father at 21.3 years of age at time of birth (at top), and the father at 52.5 years of age at time of birth (bottom). Samples highlighted in red (*n* = 9) have no blood-derived HiFi read data available. **c** Upset plot of origin assignment shows concordance between HiFi haplotypes, ONT haplotypes, and allele balance. **d** Allele balance distribution for

autosomal germline (*n* = 5145) and postzygotic (*n* = 917) SNVs across PacBio HiFi, Illumina, and Oxford Nanopore Technologies (ONT) read data. The center line defines the median value, and the box limits represent the upper and lower quartiles; the whiskers extend to the maximum and minimum points within 1.5× the interquartile range, and any points beyond are outliers. **e** Distribution of the size of autosomal insertions (*n* = 182) and deletions (*n* = 351). Source data are provided in Supplementary Data 1 and in the Source data file.

elevated TFBS mutation rate observed in testis by the Roulette model[30] (Supplementary Fig. 10).

Finally, we compared the dinucleotide mutational spectra (Fig. 2d) between germline and PZMs. We find there are 23% fewer postzygotic CpG>TpG mutations relative to germline SNVs. As expected, when comparing the same CpG site across samples with and without a DNM, we see a consistent dip in methylation in the mutated sample (Supplementary Fig. 11). Surprisingly, we notice a corresponding drop within the 10 bp of the 5′ side of de novo CpG events suggesting a potential local epigenetic effect of DNM, as has been previously observed for somatic CpG mutations[31]. In addition, PZMs are enriched for A > C and A > T substitutions (two-sided chi-squared test with Benjamini-Hochberg correction, *p*-value = $3.06 \times 10^{-4}$ and $8.05 \times 10^{-7}$, respectively) and show a marked depletion of A > G substitutions (*p*-value = $5.36 \times 10^{-5}$). Notably, we observe an expected transition to transversion ratio (Ti/Tv) of 2.10 for germline DNMs, but PZMs are significantly enriched for transversion mutations, with a Ti/Tv of 1.35 (two-sided two-proportions Z-test, *p*-value = $9.29 \times 10^{-10}$). We see no significant differences between either the germline or postzygotic single-nucleotide substitution

spectra when comparing directly between probands and their unaffected siblings (Supplementary Fig. 6c), nor did we see a significant effect of predicted ancestry on the mutation spectrum (Supplementary Fig. 12a, b).

Compared to short-read studies of autosomal SNVs for the same samples, we see a 44% increase compared to Wilfert et al. (*n* = 69 samples in both datasets), a 36% increase compared to An et al. (*n* = 60 samples), and a 30% decrease compared to Turner et al. (*n* = 17 samples)[19,20,27]. Combined, all three studies discovered 1069 DNMs that were absent from our final callset but could be lifted over to T2T-CHM13v2.0 (Supplementary Fig. 13). Approximately 70% of these DNMs were unique to the Turner callset, which we note maximized sensitivity at the expense of specificity. After applying our filtering strategy, we find that 59% and 30% of short-read-only calls appear to be inherited or false positive, respectively, leaving only 166 SNV calls supported by long reads. Of these, only 66 variants passed our full suite of filters, with exactly half excluded because they were observed in two or more unrelated samples. In total, we estimate our callset has omitted 18 false negatives from Wilfert, 8 from An, and 6 from Turner, as well as 34 events from more than one study.

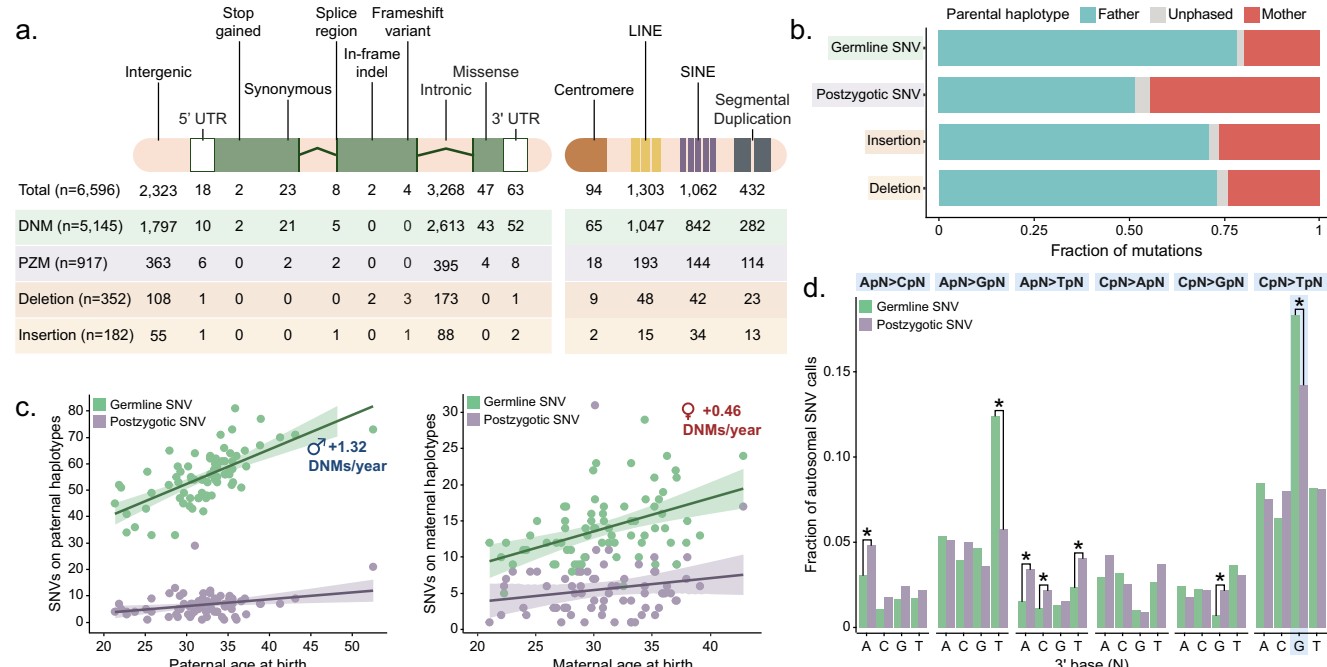

**Fig. 2 | Germline and postzygotic variant comparison. a** Functional annotations of each class of autosomal mutation, including germline (DNM) and postzygotic (PZM) SNVs, as assigned by VEP after lifting variants from T2T-CHM13 to GRCh38. **b** Fraction of autosomal SNVs and indels assigned to paternal or maternal haplotypes. **c** Number of germline SNVs assigned to each parental haplotype as a function of parental age at birth. Solid lines represent fitted linear regressions; shaded regions represent the 95% confidence interval for the expected mean number of mutations at a given parental age. **d** Dinucleotide mutation spectrum of germline and postzygotic SNVs. Benjamini-Hochberg corrected two-sided chi-squared tests were used to identify significant enrichments or depletions of postzygotic SNVs relative to germline SNVs, with a single asterisk representing $p < 0.05$. Postzygotic enrichments were observed for ApA>CpA ($p = 0.0281$), ApA>TpA ($p = 0.00179$), ApC>TpC ($p = 0.0365$), ApT>TpT ($p = 0.0214$), and CpG>GpG ($p = 4.18e-4$) mutations, and depletions were observed for ApT>GpT ($p = 1.78e-7$) and CpG>TpG ($p = 1.43e-2$) mutations. No other postzygotic enrichments or depletions rose to significance ($p > 0.05$). Source data are provided in the Source data file.

In addition, two samples from this dataset were previously examined with LRS data, yielding 195 autosomal SNVs that had been validated by ONT and HiFi reads[14]. For the same samples, we identified 196 events, 14.8% ($n = 29$) of which are unique to this study. Of those 29 SNVs, 21 are postzygotic in origin. Conversely, 28 SNVs reported in the previous study are absent from this one. More than a third ($n = 10$) were originally discovered only in GRCh38-aligned reads and were consequently not identified in our T2T-CHM13-aligned data. Of the remaining 18 variants, six were part of mutational clusters that failed our assembly-based validations, five were observed in multiple samples and excluded as errors, four failed validations by T2T-CHM13v2.0-aligned reads, and the remaining three SNVs had conflicting parental haplotype assignments. While the estimated number of DNMs for this family has remained nearly the same, this comparison reveals that we have missed approximately 16 events across these two samples, giving a false negative rate of approximately 7.5%.

**Small de novo insertions and deletions**

We applied the same criteria used for de novo substitutions to identify de novo insertion and deletions (indels, <50 bp) with one notable exception; we divided indel calls into two categories: those that were expansions or contractions of short tandem repeats (STRs) and those that were not associated with tandem repeats (TR). Most pure STR mutations were excluded from our indel callset because of the known challenges associated with accurately calling such variants with all three sequencing platforms[11]. Outside of STRs, we identified 182 and 351 de novo insertions and deletions, respectively, for an average of 7.3 indels/child on the autosomes (Supplementary Data 1), with a median indel size of ±4 bp (Fig. 1e). Just as for SNVs, indel variant counts are robust to sequence depth (Supplementary

Fig. 4b). Our callset represents a 62% increase over the four indels/child reported by Wilfert et al., a 32% increase over An et al., but a 30% decrease relative to Turner and colleagues' analysis of the same samples by Illumina sequencing. Overall, we find that 55.7% ($n = 297$) of our indel calls are unique to our HiFi callset and not previously reported in Illumina-based studies. Of the 231 calls observed in Illumina studies that were absent from our final callset, only eight appear to be truly de novo across LRS data, while 69.7% ($n = 161$) appeared inherited and the remainder ($n = 62$) were absent from LRS data.

To classify indels as germline or postzygotic in origin, we used a modified version of the same haplotype assignment pipeline. Indels, even those outside of TRs, can be noisier in LRS data, creating multiple possible alleles on a parental haplotype. As a result, indels that are truly germline in origin can appear postzygotic. To account for this noise, we determined an indel was postzygotic if there were more than three reads with an allele that differed from the called de novo event. We classified 24.2% of indels ($n = 129$) as PZMs, potentially indicating that significantly more indels are postzygotic in origin compared to the 15% of SNVs that arise postzygotically (two-sided chi-squared test, $p$-value $< 2 \times 10^{-16}$). Germline indels occur on paternal haplotypes at a 3.35:1 ratio, and indels classified as postzygotic occur on paternal haplotypes at a 2:1 ratio (two-sided Wilcoxon signed-rank test, $p$-value $= 8.81 \times 10^{-4}$). This is a more severe paternal bias than observed for SNVs and is likely due in part to misclassified germline events. There is not a significant relationship between parental age and the size of indel mutations, nor is there a significant relationship between parental age and the number of observed insertions or deletions (Supplementary Fig. 14), arguing strongly that indels within unique regions of the genome arise by distinct mechanisms.

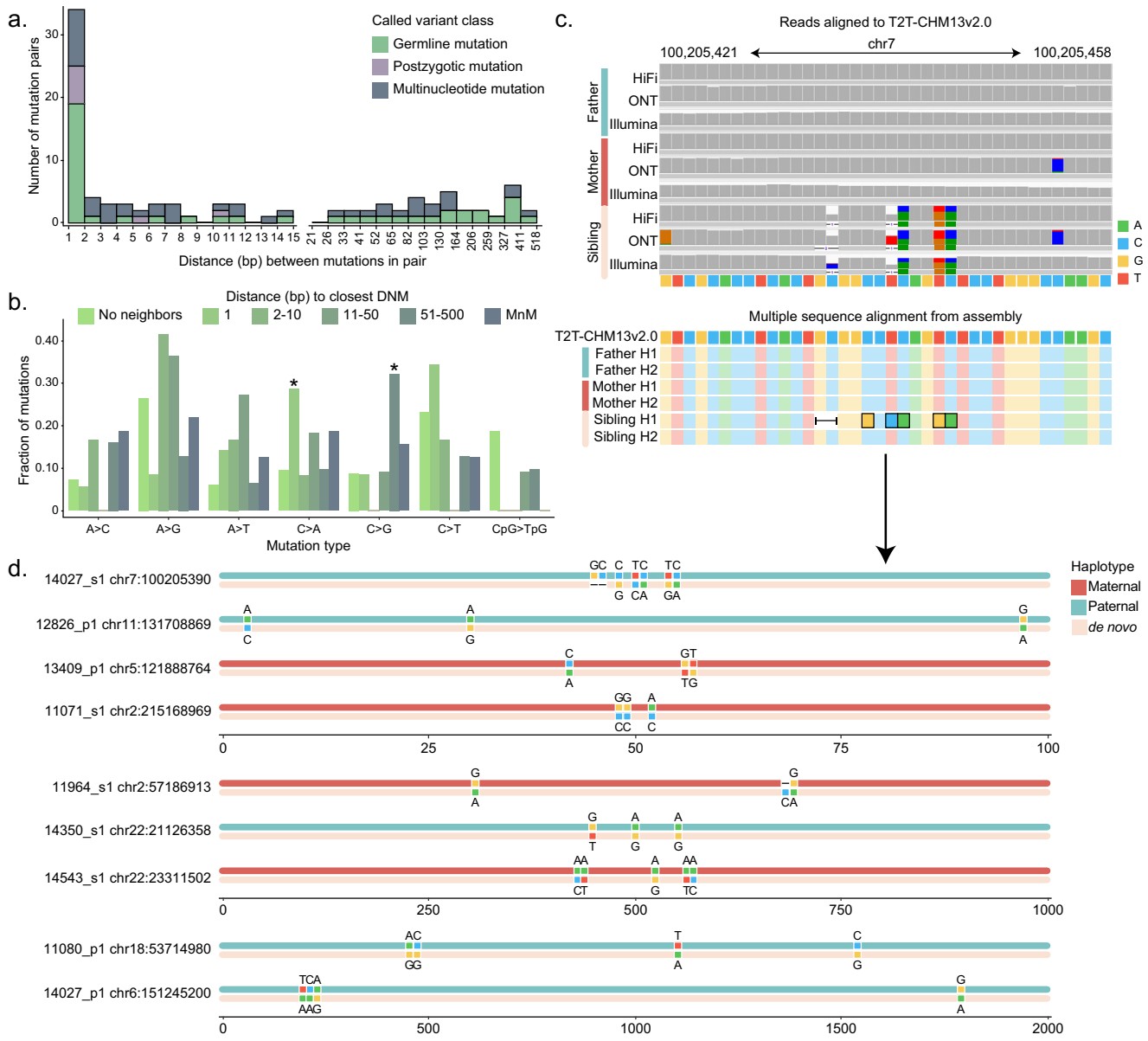

**Fig. 3 | Multinucleotide mutations. a** Distribution of distance between pairs of variants, including SNVs and indels, within 500 base pairs (bp) of each other. Variants labeled as germline and postzygotic were validated individually by confirming their presence across orthogonal read data, while multinucleotide mutations (MnMs) were also validated in combination with neighboring variants by confirming their presence in a sample's genome assembly. MnMs are all germline in origin. **b** Mutation spectrum of individual de novo mutations (DNMs) and MnMs, including paired SNV calls within 500 bp of each other. For each distance bin, Benjamini–Hochberg corrected two-sided chi-squared tests were used to evaluate the mutation spectrum relative to the spectrum of DNMs with no neighbors within 500 bp. Asterisks indicate mutation types enriched in a specific bin: C > A substitutions are significantly enriched in the 1 bp bin ($p = 0.00156$), and C > G substitutions are significantly enriched in the 51–100 bp bin ($p = 0.00162$). No other enrichments relative to unclustered mutations rose to significance ($p > 0.05$). **c** An MnM represented in sequencing reads aligned to the T2T-CHM13 reference genome, and in assembled haplotypes (H1 and H2) from all members of the trio. **d** All nine MnMs validated using assembled haplotypes. Source data are provided in the Source data file.

## Multinucleotide mutations

Early research of DNMs from parent–child family studies initially reported a small number of clustered mutations sometimes referred to as mutational "storms" or multinucleotide mutations (MnMs)[32,33]. We searched specifically in our callset for such MnMs where two DNMs occurred within 500 bp of each other. We initially identified a total of 106 mutations ($n = 103$ SNVs, $n = 3$ indels), including 23 pairs of de novo SNVs immediately adjacent to one another (Fig. 3a). However, as part of our initial filtering for DNMs we specifically excluded any mutational clusters where three or more SNVs were located within a sliding window of 1 kbp, effectively removing >1.8 million candidate DNMs from further consideration and biasing against MnMs.

We revisited those candidate events, first applying filters (Methods) that essentially exclude >99% of the calls, resulting in a total of 300 candidate single-nucleotide events. For each candidate MnM, we constructed a long-read assembly using hifiasm[28] and compared the parental and child haplotypes. Similar to read-based approaches, we considered a variant to be bona fide if it was present in a child's haplotype and absent from all four assembled parental haplotypes. Finally, we defined variant calls based on the observed mutations in the assembled data—in all cases but one the mutations concurred with the original read-based calls (Fig. 3c). This process yielded an additional 44 DNMs, including 8 unclustered SNVs and 1 unclustered indel, and 32 SNVs and 3 indels distributed across 9 clusters (Fig. 3d). There was no

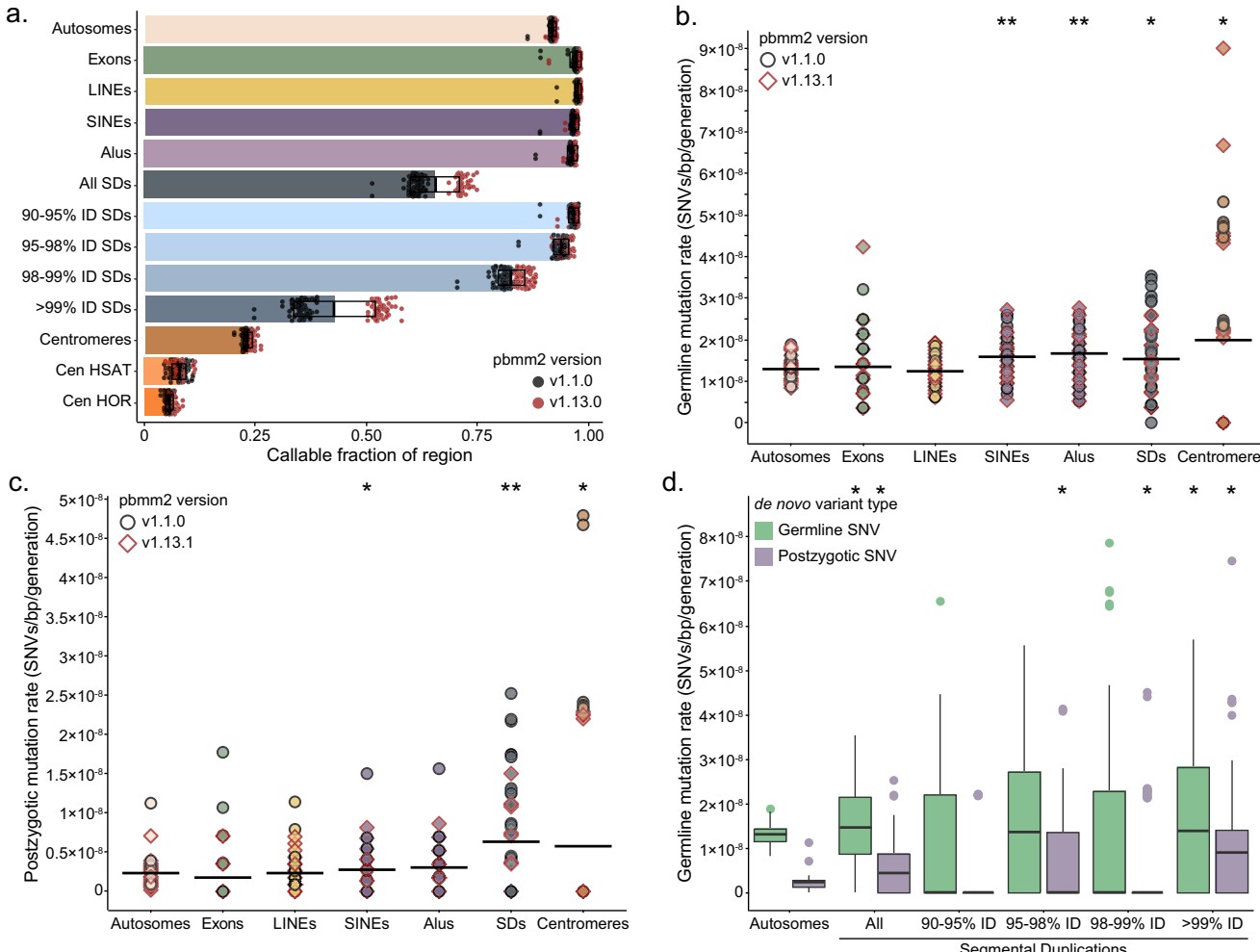

**Fig. 4 | Autosomal SNV mutation rates. a** Mean fraction of callable space in different autosomal regions, including segmental duplications (SDs), centromeric satellites (HSAT) and higher order repeats (HORs), with error bars representing 1 standard deviation. Datapoints are colored by the version of pbmm2 used to generate the HiFi alignment, either v1.1.0 (*n* = 38 samples) or v1.13.0 (*n* = 35 samples). **b** Distributions of autosomal germline DNM rate of SNVs for *n* = 73 samples across different genomic regions reveal an enrichment in Alus and SDs. Datapoints are colored by the version of pbmm2 version as in (**a**). *P*-values calculated with two-sided *t*-test and adjusted for multiple testing by Benjamin–Hochberg: exons 0.73, LINEs 0.36, SINEs 0.00025, Alus 0.000011, all SDs 0.045, centromeres 0.016. Single asterisk: *p* < 0.05, double asterisk *p* < 0.001. **c** The same analysis as in b, repeated for autosomal PZMs in all 73 samples, finds

enrichment in SDs and centromeres. Datapoints are colored by the version of pbmm2 version as in (**a**). *P*-values: exons 0.24, LINEs 0.96, SINEs 0.24, Alus 0.064, all SDs 1.91e-6, centromeres 0.037. Single asterisk: *p* < 0.05, double asterisk *p* < 0.001. **d** Autosomal germline DNM rates for all 73 samples across SDs stratified by percent identity (%ID). The center line defines the median value, and the box limits represent the upper and lower quartiles; the whiskers extend to the maximum and minimum points within 1.5× the interquartile range, and any points beyond are outliers. *P*-values were calculated with two-sided *t*-test and adjusted for multiple testing by Benjamini–Hochberg. *P*-values for germline SNVs: SDs 90–95% ID 0.38, SDs 95–98% ID 0.16, SDs 98–99% ID 0.33, SDs >99% ID 0.092. *P*-values for postzygotic SNVs: SDs 90–95% ID 0.96, SDs 95–98% ID 0.048, SDs 98–99% ID 0.037, SDs >99% ID 2.9e-5. Source data are provided in the Source data file.

parent-of-origin bias, as four MnM clusters were observed on maternal haplotypes and five were observed on paternal haplotypes. Seven of these nine clusters corresponded to repeats (4 LINEs, 1 SINE, 1 SD, and 1 low-complexity TR). Including the paired events from our SNV callset, 50% of the 128 SNVs arising in clusters associate with repetitive DNA, which represents a significant enrichment (two-sided Fisher's exact test, *p* = 0.0395) compared to unclustered mutations. Further, we find significant differences in the mutational spectra of clustered SNVs when compared to singleton DNMs (Fig. 3b). We stratified mutations into six categories based on distance to the nearest SNP and found a depletion of CpG>TpG mutations across all groups of clustered mutations. Although sample sizes are small, we observe a significant enrichment of C > A mutations in adjacent SNPs (two-sided chi-squared test with Benjamini-Hochberg correction, *p*-value = 0.0016) and an enrichment of C > G mutations in SNPs between 51 and 500 bp apart (*p*-value = 0.00016).

## DNM rate
In order to determine the DNM rate, we estimate that 91.7% (2.66/2.90 Gbp, s.d. = 23.5 Mbp) (Fig. 4a) of the human genome was assayed in this study. Because high LRS mapping quality is used to determine whether a site is callable with our method, we still excluded regions with the highest repeat sequence identity. For example, we can call variants in more than 95% of SDs with sequence identity less than 98%, whereas we can only assess 42% of SDs with over 99% identity. Our read-based approaches perform even worse in centromeres, where we can only assess approximately 6% of higher order repeats and 8% of human satellite sequence. Although we cannot fully examine the variation in these repetitive regions, we are still better equipped to study them than previous Illumina-based de novo studies, which typically exclude them completely[2,5,19,20].

Thus, limiting to high-confidence regions, we calculate a lower-bound autosomal germline mutation rate of $1.30 \times 10^{-8}$ substitutions

per base pair per generation (95% C.I. $1.27 \times 10^{-8} - 1.34 \times 10^{-8}$) (Fig. 4b) and a PZM rate of $2.30 \times 10^{-9}$ substitutions per base pair per generation (95% CI $2.16 \times 10^{-9} - 2.46 \times 10^{-9}$) (Fig. 4c), with no significant differences between proband- or sibling-specific mutation rates (Supplementary Fig. 6d). Further, we saw no significant effect of predicted ancestry on the germline or PZM rate (Supplementary Fig. 12c, d). When we restrict our analysis to GENCODEv38 exonic regions of the genome (144 Mbp of sequence, on average 97.2% callable), we find that neither the germline nor PZM rate is significantly different from the autosome-wide rate.

We also contrasted the mutation rate for different classes of repetitive DNA (Fig. 4b, d and Supplementary Fig. 15). Notably, we find a germline-specific enrichment in Alus (two-sided *t*-test with Benjamini-Hochberg correction, *p*-value = $9.93 \times 10^{-6}$) and SINEs (*p*-value = $2.35 \times 10^{-4}$). LINEs, on the other hand, show no enrichment for germline or PZMs (*p*-value = 0.34 and 0.96, respectively) relative to the autosomes. In SDs, the germline mutation rate is $1.54 \times 10^{-8}$ substitutions per base pair per generation (95% CI $1.36 \times 10^{-8} - 1.72 \times 10^{-8}$), a significant 18.5% increase over the rate across the autosomes (two-sided t-test, *p*-value = $4.26 \times 10^{-2}$). This signal is entirely driven by SDs with greater than 99% sequence identity, where the mutation rate is more than double that of the lowest identity duplications (Fig. 4d). The PZM rate is more sensitive to sequence identity and is enriched more than twofold in both SDs and centromeres (two-sided t-test, *p*-value = $1.91 \times 10^{-6}$ and *p*-value = $3.65 \times 10^{-2}$). In the case of PZMs, the strongest enrichment is in the highest identity duplications, where we observe an average of 0.85 substitutions per sample (*n* = 62).

### Sex chromosome DNM

Because of the challenges associated with ploidy differences, we used a modified version of our de novo SNV and indel discovery strategies to call variation on the sex chromosomes, treating the females (*n* = 46) and males (*n* = 27) separately. On male X chromosomes, we identified a total of 19 de novo SNVs and 5 indels, compared to the 24 SNVs and 8 indels observed on the Y chromosomes (Fig. 5a). In female samples, we found a total of 279 SNVs and 24 indels. We determined the origin of female mutations using the same haplotype strategy that we used for

the autosomes, and we classified 17.2% (*n* = 48) of female chrX mutations as postzygotic in origin. On the X chromosome, we assigned 166 and 48 germline mutations to paternal and maternal haplotypes, respectively, including 19 SNVs observed in males that must have originated in their mothers. We calculate a 3.45:1 paternal:maternal ratio on the X chromosome, completely consistent with the autosomes (two-sided two-proportion Z, *p*-value = 0.483). We estimate that we were able to discover variation in 96.1% of the female X chromosome, on average (standard deviation [s.d.] = 0.78%) (Fig. 5b). As male samples did not inherit an X chromosome from their fathers, we exclude paternal HiFi data when evaluating the male chrX. Conversely, we exclude maternal HiFi data when evaluating chrY. We are able to call on 94.8% of the male X chromosomes (s.d. = 0.80%), which is a small but significant reduction when compared to its female counterpart (two-sided t-test, *p*-value = $9.7 \times 10^{-9}$). The Y chromosome is highly repetitive, limiting our ability to call in all but 29.3% of the chromosome (s.d. = 0.75%).

We estimate the X chromosome mutation rate to be $0.46 \times 10^{-8}$ substitutions per base pair per generation in the maternal germline and $2.47 \times 10^{-8}$ substitutions per base pair per generation in the paternal germline (Fig. 5c). This enrichment of mutations in the paternal germline is even more stark on the Y chromosome, where we see a mutation rate of $5.14 \times 10^{-8}$ substitutions per base pair per generation, although given our ascertainment against repeat-rich regions this must be regarded as a lower bound. Combined with the germline mutations we observed on the autosomes, we calculate the whole-genome mutation rate in the maternal and paternal germline to be $0.52 \times 10^{-8}$ and $2.05 \times 10^{-8}$ substitutions per base pair per generation, respectively. The average whole-genome rate is $1.30 \times 10^{-8}$ substitutions per base pair per generation, which is not significantly different from the autosomal mutation rate.

### Discussion

Several studies have established that LRS increases the yield of DNM by at least 25% largely by providing better access to repeat-rich regions of the genome[11,21]. In this study of 73 children from 42 families of diverse ancestry, we identified an average of 95.3 DNM events per child, with

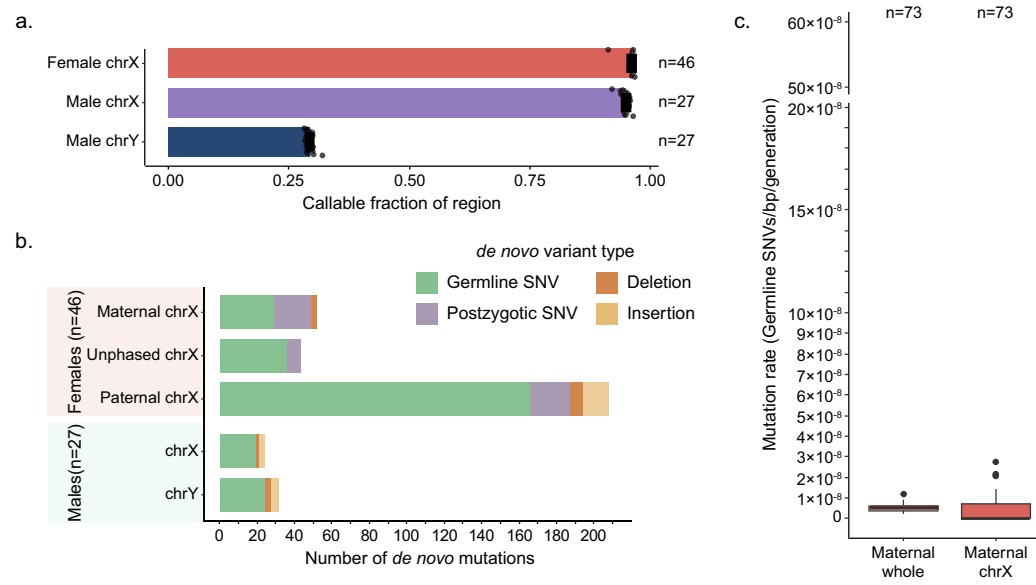

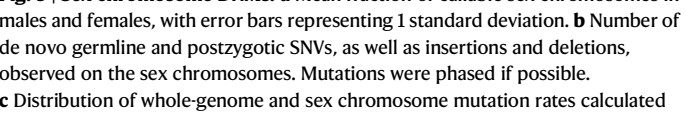

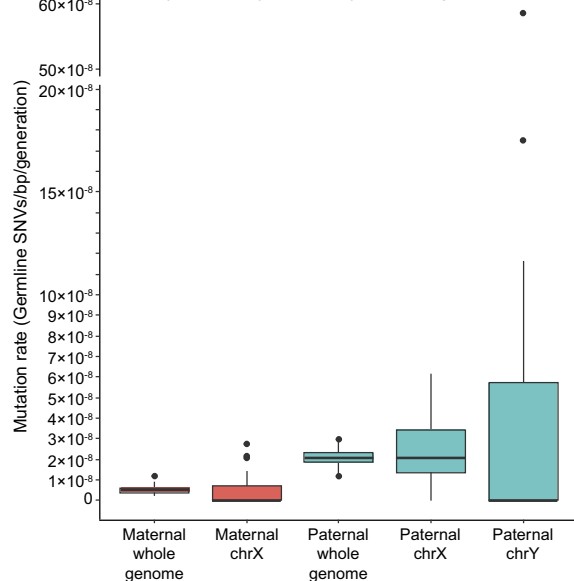

**Fig. 5 | Sex chromosome DNMs. a** Mean fraction of callable sex chromosomes in males and females, with error bars representing 1 standard deviation. **b** Number of de novo germline and postzygotic SNVs, as well as insertions and deletions, observed on the sex chromosomes. Mutations were phased if possible. **c** Distribution of whole-genome and sex chromosome mutation rates calculated

using phased germline mutations. The center line defines the median value, and the box limits represent the upper and lower quartiles; the whiskers extend to the maximum and minimum points within 1.5× the interquartile range, and any points beyond are outliers. Source data are provided in the Source data file.

**Table 1 | Comparison of DNM metrics with other studies**

| Value | Study | Estimate | CI—2.5% | CI—97.5% |
|---|---|---|---|---|
| Germline mutation rate | Sasani et al.[1] | $1.10 \times 10^{-8}$ | | |
| | Turner et al.[19] | $1.50 \times 10^{-8}$ | | |
| | Jonsson et al.[2] | $1.29 \times 10^{-8}$ | | |
| | Rahbari et al.[45] | $1.28 \times 10^{-8}$ | $1.13 \times 10^{-8}$ | $1.43 \times 10^{-8}$ |
| | Kong et al.[4] | $1.20 \times 10^{-8}$ | | |
| | current study | $1.30 \times 10^{-8}$ | $1.27 \times 10^{-8}$ | $1.34 \times 10^{-8}$ |
| Germline paternal age effect | Sasani et al.[1] | 1.44 | 1.12 | 1.77 |
| | Turner et al.[19] | 1.49 | 1.32 | 1.65 |
| | Jonsson et al.[2] | 1.51 | 1.45 | 1.57 |
| | Goldmann et al.[3] | 0.91 | 0.81 | 1.02 |
| | Rahbari et al.[45] | 2.87 | 2.11 | 3.64 |
| | Kong et al.[4] | 2.01 | standard error: 0.17 | |
| | current study | 1.32 | 0.97 | 1.67 |
| Germline maternal age effect | Sasani et al.[1] | 0.38 | 0.21 | 0.55 |
| | Jonsson et al.[2] | 0.37 | 0.32 | 0.43 |
| | Goldmann et al.[3] | 0.24 | 0.15 | 0.34 |
| | current study | 0.46 | 0.25 | 0.66 |
| Germline paternal:maternal ratio | Sasani et al.[1] | 3.96:1 | | |
| | Jonsson et al.[2] | 4.02:1 | | |
| | Goldmann et al.[3] | 3.58:1 | | |
| | Rahbari et al.[45] | 3.70:1 | | |
| | Kong et al.[4] | 3.90:1 | | |
| | current study | 3.98:1 | | |
| Germline Y chromosome mutation rate | Helgason et al.[70] | $2.30 \times 10^{-8}$ | $2.03 \times 10^{-8}$ | $2.58 \times 10^{-8}$ |
| | Xue et al.[71] | $3.0 \times 10^{-8}$ | $0.89 \times 10^{-8}$ | $7.0 \times 10^{-8}$ |
| | Kuroki et al.[72] | $4.77 \times 10^{-8}$ | | |
| | current study | $5.14 \times 10^{-8}$ | $3.26 \times 10^{-8}$ | $7.56 \times 10^{-8}$ |

A summary of commonly reported germline DNM statistics across previous short-read-based DNM studies, with results from the current study.

no significant differences between probands and their unaffected siblings, arguing against overall exposure-induced increases in mutagenesis as an underlying epidemiological cause in at least these unsolved autism families. While this is not the first study of DNMs leveraging both long- and short-read data[11,21], this larger and more diverse set of samples has yielded more DNM and PZM calls than previous long-read studies. Thus, it provides detailed insights regarding de novo genome-wide patterns of variation, including: (1) mutation rate variation between different classes of repeats, including a significant enrichment in SINE repeat elements not observed in LINEs; (2) mutation enrichment in SDs, a feature directly correlated with higher sequence identity; (3) postzygotic enrichment in SDs and repetitive regions; and (4) MnMs that could only be validated with the availability of near-complete genome assemblies. In total, our callset represents a 20–40% increase in DNM discovery per sample when compared to earlier short-read studies of the same samples[19,20,27]. Our estimate of the number of autosomal SNVs assigned to the germline (83 DNMs/sample) is similar to other reports using LRS (74.5–86) (Tables 1 and 2)[25,34]. Notwithstanding, the number of DNMs per child is significantly less than the 153 DNMs per generation recently reported by Porubsky et al. That study increased sensitivity by applying additional sequencing platforms (Aviti), better equipped to discover indel mutations within homopolymers or STRs, as well as assembly-based approaches to recover DNMs in the most complex regions of the human genome. Thus, while we have increased DNM sensitivity, we still regard our mutation rates as a lower bound, significantly underestimating the indel mutation rate, especially in STRs and overall in more complex regions of the genome.

Another major advantage of LRS is the ability to phase variants, assigning them unambiguously to parental haplotypes. Without pedigree information, short-read data can typically be used to phase up to 20% of DNM calls[2,3], but we were able to phase 97.7% and 97.0% of autosomal SNVs and indels, respectively. In addition, the phasing data can be used to classify DNMs arising in the germline versus the early embryo. In our study, we estimate that 15.1% of SNVs (12.6 DNMs per child) are postzygotic in origin, almost doubling earlier estimates of 4–10%[1,14,16,17]. This enrichment may be in part the result of somatic mutations that arose and clonally expanded in blood; such mutations cannot be filtered without access to another tissue for sequencing, a clear shortcoming of our ability to generate a highly specific PZM callset. However, blood data were collected when our samples were under the age of 18, so we do not expect to see as much somatic contamination as we would in an adult dataset[35–37], and there is no correlation between sample age and total PZM count (Supplementary Fig. 5). Further, over 98% of our PZMs have HiFi AB greater than 0.1, indicating their presence in at least 20% of sequenced cells—at this threshold, detection of somatic variants, even those that may have clonally expanded during early childhood, is unlikely[38]. Finally, this higher estimate is consistent with our recent multigenerational LRS sequence assembly analysis where we showed that ~60% of PZMs called with this method are transmitted to the next generation[6]. That analysis confirms that a significant fraction of such PZMs are present in the germline (in addition to the blood) and suggests at least two-thirds of the PZMs discovered here are likely early embryonic mutations.

PZMs show unique mutational signatures and biases compared to germline mutations, likely due to the different environment in which

**Table 2 | Parental age-adjusted mutation rates across studies**

| Data Type | Study | Autosomal SNVs/sample | Per generation rate | Average parental age | Per year rate | Rate for 30-year-old parents |
|-----------|-------|------------------------|---------------------|----------------------|---------------|------------------------------|
| Short-read | Kong et al.[4] | 63.2 | $1.20 \times 10^{-8}$ | 29.7 | $4.04 \times 10^{-10}$ | $1.21 \times 10^{-8}$ |
| | Rahbari et al.[45] | 64 | $1.28 \times 10^{-8}$ | 29.8 | $4.30 \times 10^{-10}$ | $1.29 \times 10^{-8}$ |
| | Jonsson et al.[2] | 70.3 | $1.29 \times 10^{-8}$ | 30.1 | $4.24 \times 10^{-10}$ | $1.27 \times 10^{-8}$ |
| | Sasani et al.[1] | 70.1 | $1.10 \times 10^{-8}$ | 27.6 | $3.99 \times 10^{-10}$ | $1.20 \times 10^{-8}$ |
| Long-read | Noyes et al.[21] | 86 | $1.41 \times 10^{-8}$ | 35.4 | $3.99 \times 10^{-10}$ | $1.20 \times 10^{-8}$ |
| | Kucuk et al.[25] | 75 | | 36 | | |
| | Ng et al.[34] | 82.5 | | | | |
| | Porubsky et al.[11] | 74.5 | $1.17 \times 10^{-8}$ | 27.3 | $4.33 \times 10^{-10}$ | $1.30 \times 10^{-8}$ |
| | current study | 83 | $1.31 \times 10^{-8}$ | 31.4 | $4.16 \times 10^{-10}$ | $1.25 \times 10^{-8}$ |

A summary of reported SNV counts and mutation rates across short- and long-read-based DNM studies. When parental age data were available, we adjusted the reported per-generation mutation rate by dividing by the average age of parents in the study, following the precedent of Jonsson et al.[2].

they arose. For example, PZMs are depleted for CpG-associated mutations and are enriched for transversions when compared to those classified as germline (postzygotic Ti/Tv of 1.3 vs. germline Ti/Tv of 2.1). Across published PZM callsets, there is not a consensus on the PZM spectrum, as Jonsson et al. report enrichments of A > C, C > A, and C > T mutations, while Sasani et al. report only an A > T enrichment, and Porubsky et al. report both an A > T enrichment and CpG>TpG depletion[1,11,12]. In this study, we replicate the postzygotic enrichment of A > C and A > T mutations, as well as the depletion of CpG mutations. We find that 80% (3.95:1) of germline mutations occur in the paternal germline, where an additional 1.3 SNVs arise with every passing year, in contrast to PZMs, which are more equally distributed among parental haplotypes (53% of PZMs arise on paternal haplotypes). This constitutes a modest but significant paternal bias for PZMs (1.15:1), consistent with previous studies of human PZMs that also observed some evidence of paternal bias but did not rise to significance (Porubsky 1.17:1, Sasani 1.08:1)[1,11]. Further, Lindsay et al. described a significant paternal bias in 55 mouse PZMs that was notably absent from humans[10], although they only characterized a total of 25 human PZMs, and work from Harland et al. identified a depletion on paternal haplotypes when examining 79 mosaic mutations in cattle[39]. Because mosaic PZMs have definitionally low AB and are subject to greater sampling error than germline events, it is not surprising to see disagreement across studies with low sample sizes. Larger studies will be required to determine what, if any, effect parental haplotype has on the mutational landscape of the early embryo. Paired with a slight paternal age effect of 0.26 PZMs per year, this paternal bias could suggest a PZM mechanism involving embryonic repair of lesions that arise in the paternal germline as a father ages[40,41]. One possibility may be single-stranded DNA damage exists as heteroduplex in the sperm that cannot be adequately repaired until after conception. Even after fertilization, DNA transcription does not begin until the 4–8 cell stage[42], so some lesions may persist, constrained to the repair machinery available in the oocyte[43]. Coupled with error-prone early cell divisions[9,44], the embryo may preferentially turn to repair mechanisms like allelic and interlocus gene conversion (IGC) to correct errors. Another possibility is that some of these mutations may in fact be germline in origin, but a postzygotic gene conversion event rescued the original allele in a subset of daughter cells, resulting in three haplotypes. We would expect to see such a pattern in high-identity regions such as SDs, and it may help to explain the observed paternal age effect.

Based on the 91% of the callable genome, we calculate a germline mutation rate of $1.30 \times 10^{-8}$ substitutions per base pair per generation. Excluding PZMs, we predict the mutation rate for 30-year-old parents to be $1.25 \times 10^{-8}$, which is very close, on first blush, to estimates of $1.21–1.30 \times 10^{-8}$ made by SRS- and LRS-based studies[1,2,4,11,34,45] (Tables 1 and 2). We estimate, however, that PZMs contribute an

additional $0.23 \times 10^{-8}$ substitutions per base pair per generation for a combined germline and PZM DNM rate of $1.53 \times 10^{-8}$—a marked increase over short-read-based estimates. Given that nearly 5% of our total callset is composed of PZMs with AB > 0.30, a standard filtering threshold in DNM studies, the true germline mutation rates in these previous studies could be overestimated by as much as 5%.

Consistent with our study of a multigenerational family[11], the mutation rate is elevated in repetitive DNA but not uniformly so. We find significant increases in DNM for SINEs and SDs but not LINE repeats, perhaps due to the decreased CG content for the latter and the suspected role of IGC in elevating the mutation rate of these regions[46–48]. Both germline mutations and PZMs are significantly enriched in SDs, and the acceleration increases the longer and more identical the repeats become (Fig. 4c). Specifically, the germline mutation rate for SDs with >99% sequence identity is 30% higher while the postzygotic rate is over fourfold higher. These findings suggest that SDs are particularly prone to single-nucleotide substitutions early in embryogenesis. Although the strict germline mutation rate increase (18.5%) is more modest than we might expect based on previous population genetic estimates (which predicted an increase of 60%)[48], if we combine both the germline and postzygotic rate, we estimate an overall 42% increase in mutation over SDs. The fact that DNMs in SDs are significantly depleted in CpG substitutions and show an excess of transversions once again points to a role for IGC in driving some of this acceleration[48,49]. In addition, these signatures are also consistent with the types of PZMs observed in SDs.

Another potential result of IGC and repeat-associated mutation is clustered de novo MnMs. We identified nine such MnM clusters that were supported by both read-based and assembly alignments. Six of the mutation clusters mapped to repeats, including LINEs (n = 4), SINEs (n = 1), and SDs (n = 1). We hypothesize that some of these MnMs are likely the result of faulty repair with polymerase ζ, which preferentially creates paired single-nucleotide substitutions, and in fact 9 of our 23 tandem SNV mutations are GC > AA or GG > TT substitutions, a hallmark of this polymerase[33,50]. Notably, four of our MnM clusters have an exact sequence match to the de novo allele in a paralogous repeat mapping elsewhere in a parental genome, suggesting these sequences served as the original template for the IGC event[51]. Comparatively, only 6% of germline SNVs and 8% of postzygotic SNVs are gene conversion candidates. The plurality of conversion candidates are in LINEs (44%), with an additional 20% each in Alus and SDs. While it is likely that many conversion candidates did not actually arise from IGC events, IGC clearly contributes to the mutational landscape of these repetitive regions.

Despite the increased mutation rates we report, additional DNMs remain to be discovered. While we were able to assess small variants in approximately 91% of the genome, the remaining 9% is among the most mutable[11]. Owing to the unreliability of Illumina and LRS-based

variant calls, for example, we excluded some of the most complex TR insertions and deletions within STRs and VNTRs from this analysis. We only analyzed approximately 6% of centromeric alpha-satellite DNA and excluded over three-quarters of the Y chromosome, where mutation rates are known to be more than an order of magnitude higher[11,52]. We also did not assess large-scale structural variation because most de novo SVs map to complex regions ill-suited for characterization by read-based alignments[11]. Accessing these regions and these particular classes of mutation will require assembly-based approaches which, in turn, require sequencing of higher molecular weight DNA, not as readily derived from retrospective clinical research material. A dedicated effort to sequence genomes telomere-to-telomere from many more diverse families should be undertaken to better understand patterns, rates, and biases of DNM among humans.

## Methods

### Illumina sequencing
We used previously published Illumina data generated by the New York Genome Center for the SSC and SAGE samples[19,20]. Briefly, each sample was sequenced on the Illumina X Ten platform using 1 μg of blood-derived DNA with an Illumina PCR-free library protocol.

### HiFi sequencing
We used the same blood and lymphoblastoid cell-line-derived HiFi sequencing data as described in Sui et al[28]. Briefly, DNA was extracted according to the manufacturer's recommendations and sequenced on either the PacBio Sequel II or Revio platform.

### ONT sequencing
We used the same ONT data as described in Sui et al.[28]. Briefly, ONT data were generated from DNA extracted from lymphoblastoid cell lines using a modified Gentra Puregene protocol. Libraries were constructed using the Ligation Sequencing Kit (ONT, SQK-LSK110) with modifications to the manufacturer's protocol. The libraries were loaded onto a primed FLO-PRO002 R9.4.1 flow cell for sequencing on the PromethION, with two nuclease washes and reloads after 24 and 48 h of sequencing.

### Alignment to the reference genome
We selected T2T-CHM13v2.0 as our reference genome, as it allows us to align reads to repetitive regions that were not represented in the previous reference. For all females, we masked the Y chromosome from the reference before alignment, and for all males, we masked the pseudoautosomal regions of the Y, following the guidance from Rhie et al.[53] Illumina data were aligned using BWA-MEM v0.7.17[54]. We aligned both HiFi and ONT data using minimap2[55], with the help of the pbmm2 v1.13.1 (https://github.com/PacificBiosciences/pbmm2) wrapper for handling the HiFi data. It is important to note that we aligned the first batch of processed samples ($n = 42$) using pbmm2v1.1.0 and the second batch of samples ($n = 73$) using pbmm2v.1.13.0. The newer version of pbmm2 performed notably better in higher sequence identity regions, but there was no difference in the average number of DNM calls found across both batches of samples.

### de novo SNV discovery and validation on the autosomes
Variant calling was performed using aligned HiFi data and two variant callers, GATK HaplotypeCaller v4.3.0.0[56] and DeepVariant v1.4.0[57], following the same filtering strategy outlined in Noyes et al.[21]. For each caller, we naively identified candidate de novo events by selecting any variant where both parents were homozygous for the reference allele and the child had at least one alternate allele. We took the union of both candidate de novo callsets, retaining only variant calls where the child's genotype quality was at least 20, resulting in an initial callset of 2,159,552 SNVs across all 73 samples. To eliminate runs of candidate de

novo events that were actually the result of a dropped haplotype in one parent, we eliminated any regions where three or more SNVs were found in a sliding 1 kbp window, removing a total of 1,837,202 candidate events from our callset.

Next, we examined the HiFi, ONT, and Illumina reads that spanned each candidate variant in a child and both parents. For a child's HiFi read to be considered, it had to be derived from blood data, but both parental blood and cell line reads were retained (all ONT data were derived from cell lines; all Illumina data from blood). Long reads with mapping quality <59 were excluded (we did not filter short reads on the basis of mapping quality). We partitioned reads into three categories based on the base quality (probability that a base was correctly called) at the site of the variant: reads with base quality >20 (high quality), reads with base quality between 10 and 20 (low quality), and reads with quality <10, which were discarded. For each sequencing platform, we counted the number of reads that supported the reference and alternate alleles in both parents and the child and used them to determine whether a variant was truly de novo or inherited. For HiFi and Illumina data, we required that each parent have fewer than one high-quality or two low-quality reads with the de novo allele, and the child has at least one read with the de novo allele. Since ONT is slightly less accurate, we required fewer than two high-quality or three low-quality reads with the de novo allele. Once each variant was examined in each platform, we combined the validations, determining that a variant was inherited if it looked inherited in at least one platform, and truly de novo if it was supported in at least two platforms. Across all samples, 14,187 candidate mutations passed this initial round of filtering.

We returned to the aligned HiFi reads for every sample in the dataset (parents and children), checking every candidate de novo allele to see if it was represented in more than one sample, removing approximately 7000 variants that we determined to be recurrent errors. If a variant was not present in a TR, we required that it be unique to the child it was identified in, and if the variant was in a TR, we allowed it to be observed in one unrelated sample. Although we evaluated the frequency of each variant in gnomAD4.1 (Supplementary Data 5), we did not use population frequency as a filter as we wanted to specifically exclude sequencing and alignment artifacts arising from mapping long-read data to T2T-CHM13v2.0. In addition, variants in TRs had to have an average AB greater than 0.05 across all platforms. We then removed low-quality variants that had dubious support across two or more platforms (typically noisy parental data with alternate alleles different from the de novo allele). We excluded variants in regions flagged by RepeatMasker that failed AB filters (0.1 if it was also in a TR, 0.08 if not)[58]. Lastly, we removed variants in or adjacent to homopolymers that involved the homopolymer subunit (i.e., an A to T substitution on the edge of an A homopolymer), as those variants are typically sequencing artifacts and difficult to validate across all sequencing platforms. We validated a total of 6070 variants that we then assigned to haplotypes. A final 40 variants could not be uniquely assigned to one parent and were excluded, resulting in a final callset of 6030 autosomal SNVs.

### de novo indel discovery and validation on the autosomes
We generated candidate indel callsets using the same combined GATK and DeepVariant callset, naively selecting for any insertions or deletions present in a child and absent from its parents. We divided these calls into two categories: TR mutations—where one or more subunits were added or subtracted, and indels—that either did not involve a perfect TR motif or did not overlap with a TR ($n = 616,873$). We applied a similar read-based validation strategy for indels as we did for SNVs. We required that HiFi, ONT, and Illumina reads have mapping quality of 60 (the highest possible) and that they fully span the variant site, with at least 10 bp of flanking sequence before and after, to ensure that we captured the full allele.

We excluded variants in TRs. To validate indels outside of TRs, we examined child and parental data across all three sequencing platforms, counting the number of de novo alleles. If a child had a sibling in our dataset, we also examined the sibling's read data. We considered a variant to be inherited if one parent or the sibling had a read supporting the de novo allele in any platform. We deemed a variant to be truly de novo if we observed the de novo allele in the child in both HiFi and Illumina data, resulting in a total callset of 596 de novo indels. Finally, we visually inspected every indel call in IGV, removing any indels with several alternate alleles. Many of these sites were located in homopolymers or repetitive sequence, resulting in a diversity of alleles and often obscuring the inheritance pattern, resulting in false positive calls. After manual inspection, our final callset was composed of 533 indels.

## Multinucleotide mutation (MnM) discovery

To identify MnMs, we began with the excluded 1,837,202 calls from the SNV validation pipeline. We removed the clustered mutation filter and applied the same validation steps as for unclustered SNVs. Only 15,713 mutations remained after the initial three-platform validation, and 300 passed every filter. We phased these 300 SNVs, excluding 10% ($n = 30$) because they were found on both parental haplotypes. For the remaining 270 SNVs, we examined parent and child assemblies generated with hifiasm[28,59].

We defined MnM regions by examining GATK VCFs, selecting any SNVs within a 5 kbp sliding window around the validated event. We then subset these regions from parent and child assemblies, only retaining variants when all four parental haplotypes and both child haplotypes were fully assembled in the 50 kbp surrounding the clustered event. We used MAFFT[60] to generate a multiple sequence alignment of all six assembled sequences, as well as the corresponding sequence from T2T-CHM13v2.0. We retained any variants that were unique to the child's haplotype and excluded cases in which the surrounding sequence had excessive mismatches surrounding the region, a phenomenon we commonly observed in misaligned or misassembled regions such as centromeres. This process yielded 44 DNMs, 9 unclustered events (8 SNVs and 1 indel), and 9 clustered events. In cases where the assembly-based variants disagreed with the GATK-called variants, we deferred to the assembly-based results (Fig. 3c).

## Sex chromosome de novo discovery

To identify variation, we used ploidy-aware GATK HaplotypeCaller v4.3.0.0, treating the female chromosome X as diploid, and the male sex chromosomes as haploid. Females and males from each family were jointly genotyped separately, and variant calls were filtered using the same parameters as autosomal variants.

For female children, we naively identified de novo variation by selecting sites that were homozygous reference in the mother and hemizygous reference in the father, and the child had an alternate allele, identifying 29,195 SNVs and 17,772 indels. We excluded 21,838 SNVs that were in clusters of three or more within 1 kbp, then used most of the SNV filtering strategy that we applied to autosomal variants, examining HiFi, ONT, and Illumina reads to ensure each variant was unique to the child in which it was identified. We used the same filtering parameters for sites in TRs but did not filter based on RepeatMasker or homopolymer annotations[58], as few sites were in such regions. In total, 283 SNVs on female X chromosomes passed our validations. After assigning these variants to parental haplotypes, four had conflicting parental information and were excluded from our final callset of 279 SNVs. We used the same autosomal indel validation pipeline for non-TR variants, resulting in a final callset of 26 indels on female X chromosomes.

For male children, we treated X and Y chromosome variation separately, excluding the pseudoautosomal regions on both. We naively identified variants on the X chromosome by comparing a child to his mother, selecting any sites where the child had a different allele

(not required to be reference or alternate). Conversely, we selected sites on the Y chromosome where a child had a different allele than his father. We identified 21,107 SNVs and 78,939 indels on male X chromosomes, and 100,225 SNVs and 8215 indels on male Y chromosomes. To validate SNV calls on the male X chromosome, we applied the same filtering strategy as the female X, first evaluating HiFi, ONT, and Illumina sequencing data to ensure that a variant was unique to a child, and then filtering sites in TRs. We applied an additional male-specific filter, requiring the AB for SNVs to be 1, as we would expect a true de novo event to be present on all reads deriving from an X or Y chromosome. This more stringent filter helped to eliminate variants resulting from mismapping between the sex chromosomes and yielded a total of 19 SNVs on male X chromosomes. For SNV calls on the Y chromosome, we simply checked a child and the father's sequencing data across all three platforms to ensure that the variant was not present in the father, resulting in 140 SNVs. We applied our AB = 1 filter and manually validated all 140 SNVs in IGV and excluded any that had low sequencing depth (fewer than 3 reads) and extensive local variation, resulting in a final callset of 24 SNVs on male Y chromosomes. Please note that, as a result of our stringent AB filter, we were not able to evaluate postzygotic variants on the male sex chromosomes, as they would necessarily have AB < 1. We used the same indel filtering strategy for male sex chromosome variants as for female, except we only examined maternal sequencing data for X and paternal data for Y chromosome variants. In total, we found three indels on male X and eight indels on male Y chromosomes.

## Phasing

For every DNM, we identified informative SNPs within an 80 kbp window centered at the mutation site based on variant calls from our GATK4 run using HiFi data. An informative SNP is defined as any SNP whose origin can unambiguously be assigned to one parent: for example, a site where one parent is 0/0, the other parent is 0/1 or 1/1 and their child is 0/1. We then examined the HiFi read data for the child, examining every read derived from blood DNA that passed our read filters (mapping quality ≥59 and base quality ≥20 at the site of the DNM). We assigned each read to a maternal or paternal haplotype by calculating an inheritance score based on the presence of tagging SNPs. We gave tagging SNPs a value of ±1 depending on whether they were inherited from the mother or father, and then took the average of these values, inversely weighted by each SNP's distance from the DNM site. A negative inheritance score indicated a paternally inherited read, and a positive score indicated a maternally inherited read. If all the reads with the de novo allele could be assigned to one parent, we assigned them as the parent-of-origin. If the de novo allele was present on both paternal and maternal haplotypes, it was left unphased. Using this method, we were able to phase 90.7% of all SNVs ($n = 5509$), while the remaining 561 had no tagging SNPs or ambiguous parental data.

DNM origin was also evaluated using ONT reads. We applied the same read-filtering steps, with the caveat that all ONT data were derived from cell lines, and calculated inheritance scores based on informative SNPs. For each SNP, we compared the HiFi and ONT haplotype assignments, preferentially selecting the HiFi assignment in cases of disagreement ($n = 8$) between the two sequencing platforms. With the ONT data, we were able to phase an additional 417 DNMs, leaving just 3% of our DNMs ($n = 136$) unphased. In cases where tagging SNPs were present, but neither ONT nor HiFi data could unambiguously phase a de novo variant ($n = 48$), we assumed the variant was due to sequencing error and excluded it from the final DNM callset.

## Assessment of postzygotic mutation

Using the filtered and parentally assigned HiFi and ONT reads from our phasing pipeline, we counted the number of reference and alternate alleles derived from each parent and determined that a DNM was postzygotic in origin if we detected at least two reads with the

reference and one read with the alternate allele assigned to one parent's haplotype.

We also predicted whether a mutation was likely to be germline or postzygotic based on the new variant's AB across HiFi, ONT, and Illumina sequencing data. First, we filtered reads according to the same parameters used in the phasing pipeline, restricting HiFi data to only blood-derived reads, and counted the number of reads with the reference and alternate alleles. We used these counts to calculate AB for each platform and then used a chi-squared test (using chi2_contingency from the Python package scipy.stats) to determine whether the three AB values were concordant. In cases where the AB was not concordant, we could not confidently predict whether the variant was postzygotic, so it was supposed to be germline. If the AB was concordant across platforms, we pooled the reference and alternate allele counts and tested whether the total AB was significantly less than 0.5 using a binomial test (binomtest from scipy.stats). Any variants with significantly low AB were predicted to be postzygotic in origin.

To make the final determination of mutation origin, we combined results from the HiFi and ONT haplotypes and AB-based predictions. In cases where HiFi and ONT haplotypes disagreed ($n = 286$), we used the origin assignment that matched our AB prediction, and in cases where HiFi and ONT haplotypes were ambiguous ($n = 184$), we used the AB prediction. In total, we determined that 917 de novo SNVs were likely postzygotic in origin, and 5145 arose in the parental germline.

### Functional annotation of DNMs

Variant consequences were annotated on both T2T-CHM13v2.0 and GRCh38 references using UCSC LiftOver[61] and the Ensembl Variant Effect Predictor (VEP, v111.0 and v110.1)[29]. DNMs overlapping with NDD genes (1,238 SFARI genes, 664 NDD genes from Fu et al.[26], 615 NDD genes from Wang et al.[62], and 102 ASD genes from Satterstrom et al.[63]) are flagged in Supplementary Data 2. We further incorporated gene-level constraint metrics (pLI and LOEUF) from gnomAD v4.1.

Variant-level functional and pathogenicity annotations, including ClinVar clinical significance, SIFT, REVEL, AlphaMissense phred, and AlphaMissense scores from dbNSFP v4.8a[64], EVE and popEVE[65], and MisFit[66], were added using GRCh38 coordinates. Noncoding features were annotated with the comREG pipeline (https://github.com/EichlerLab/asap) described in Sui et al.[28], incorporating the genomic Gnocchi score[67], regulatory regions (from UCSC and the developing brain[68]), and repetitive regions. Additionally, using the transmission curation pipeline described in Sui et al.[28], we identified 25 de novo SVs (including TR expansions and contractions) in the 20 probands and 16 in the 13 unaffected siblings (Supplementary Data 3). We then quantified the genomic distance between DNMs and these de novo SVs using BEDTools closest.

### Callable genome and mutation rate calculation

To determine where we were able to identify de novo variation in the genome, we assessed HiFi data for every trio. We first used GATK HaplotypeCaller v4.3.0.0 with the option "ERC BP_RESOLUTION" in order to generate a genotype call at every site in the genome. Only sites where both parents were genotyped as homozygous reference (0/0) were considered callable, as sites with a parental alternate allele were excluded from our de novo discovery pipeline. We then examined the HiFi reads from a sample and its parents, restricting to only primary alignments with mapping quality of at least 59. For children with blood data available, we only considered HiFi reads derived from blood, but we considered both blood and cell line data for parents. We counted the number of reads with a base of at least a quality score of 20 at every site in the genome and then combined this information with our variant calls. A site was deemed callable if both parents and the child each had at least one high-quality read with a high-quality base call. We observed an average of 2.66 Gbp (out of 2.90 Gbp, s.d. = 24.9 Mbp) such sites across the autosomes. For female children, the callable

chromosome X was determined the same way, whereas for the male children, we only considered the mother's HiFi data when examining the X chromosome and the father's HiFi data when examining the Y chromosome. In addition, male sex chromosomes were not restricted to sites where both parents were genotyped as reference—each parent was allowed to carry an alternate allele.

It is important to note that we processed our HiFi data in two separate batches, aligning them with two versions of pbmm2 (v1.1.0 and v.1.13.1). Due to improvements in alignment quality over repetitive regions, samples aligned with the newer version have an average of 24 Mbp of additional callable space across the autosomes, with the most notable gains in high-identity SDs. We did not observe an increase in DNM calls in samples aligned with the newer version of pbmm2.

We calculated the germline autosomal mutation rate for every sample by dividing the number of germline autosomal DNMs by twice the number of base pairs we determined to be callable. For PZMs, we used the same denominator. In females, the amount of callable sex chromosomes was defined as twice the number of callable bases on the X chromosome, and in males it was defined as the sum of the callable bases on the X and Y chromosomes. For each feature-specific mutation rate (such as SDs), we intersected both a sample's de novo SNVs and the sample's callable regions with coordinates of the relevant feature. We then calculated the mutation rate by dividing the number of SNVs in the region by the amount of callable space in the region.

### Ethics

Ethical approval for this study was granted by the University of Washington IRB Committee B, under STUDY ID: STUDY00000383.

### Reporting summary

Further information on research design is available in the Nature Portfolio Reporting Summary linked to this article.

## Data availability

The data used for analysis, including underlying sequence data, assemblies, and alignment files, are available to approved researchers in the SFARI Base under the accession number SFARI_DS0000104 (https://base.sfari.org/dataset/DS0000104) and through the National Institute of Mental Health Data Archive (NDA) under Collection ID 3780. Source data are provided with this paper.

## Code availability

Code and scripts used for the analyses presented in this manuscript are available in GitHub at https://github.com/mdnoyes/denovo_calling[69]. Any additional information required to reanalyze the data reported in this work paper is available from the lead contact upon request.

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

## Acknowledgements

This work was supported, in part, by the US National Institutes of Health (NIH R01MH101221 to E.E.E.) and the Simons Foundation (SFARI #810018EE to E.E.E.). E.E.E. is an investigator of the Howard Hughes Medical Institute. We thank Tonia Brown for assistance in editing this manuscript. This article is subject to HHMI's Open Access to Publications policy. HHMI lab heads have previously granted a nonexclusive CC BY 4.0 license to the public and a sublicensable license to HHMI in their research articles. Pursuant to those licenses, the author-accepted manuscript of this article can be made freely available under a CC BY 4.0 license immediately upon publication.

## Author contributions

E.E.E. and M.D.N. conceptualized the study. K.M.M., K.H., J. Kordosky, G.H.G., J. Knuth, and A.P.L. generated the data. Y.S., Y.K., I.W., and N.K. performed data quality control. M.D.N. and Y.S. conducted the formal analysis. M.D.N. created the visualizations. M.D.N. developed the methodology. M.D.N. wrote the original draft. M.D.N., Y.S., and E.E.E. reviewed and edited the manuscript. E.E.E. supervised the study.

## Competing interests

E.E.E. is a scientific advisory board (SAB) member of Variant Bio, Inc. All other authors declare no competing interests.
