## [Transparent Peer Review file · Nature Communications]

Long-read sequencing of families reveals increased germline and postzygotic mutation rates in repetitive DNA

Corresponding Author: Professor Evan Eichler

Version 0:

Reviewer comments:

Reviewer #1

(Remarks to the Author)

Noyes et al apply long- and short-read sequencing to identify germline and postzygotic mutations across 42 families with a child diagnosed with autism spectrum disorder (ASD). The authors find a higher rate of de novo mutations (DNMs) than previously reported, largely due to the increased sensitivity of long-read sequencing (LRS). Notably, they are able to phase DNMs using LRS, which also allows them to distinguish postzygotic mutations (PZMs) from germline mutations. The authors also investigate repetitive regions of the genome, including segmental duplications and retrotransposable elements, which are difficult to assess via short-read sequencing approaches and indeed reveal a higher rate of DNMs. Overall, this study provides valuable insight into DNMs and PZMs, and particularly highlights the utility of LRS in detecting variants in “difficult” regions of the genome.

However, several aspects of the study require clarification and further contextualization. In particular, the interpretation of PZM parental bias needs reconciliation with previous findings, the novelty relative to the authors’ own previous studies should be made explicit, figure presentation should be improved for clarity, and additional subgroup analyses may uncover biologically meaningful patterns.

Specific Comments:

1. The authors should explicitly specify the novelty of this work compared to the authors’ previous study (Porubsky et al., Nature 2025, ref#6), as many of the core findings (e.g., rates of DNMs in repetitive DNA, proportion of postzygotic events) are similar. Without an ASD-specific comparison (e.g., probands vs. unaffected siblings), the novelty of this manuscript is limited.
2. The modest paternal bias of PZMs is difficult to interpret biologically, as PZMs should occur equally on maternal and paternal alleles. Could this reflect misclassification of some germline variants as postzygotic? For indels, they commented that “a more severe paternal bias than observed for SNVs and is likely due in part to misclassified germline events”. Why does it not apply to SNVs? Consider reanalyzing PZMs restricted to those with strong SRS support (assuming SRS data have higher depth for improved classification). Discuss whether read depth influences PZM calls.
3. The authors’ previous study (Porubsky et al., Nature 2025, ref #6) reported that 16% of de novo single-nucleotide variants are postzygotic, with no paternal bias, including early germline mosaic events. Please reconcile this discrepancy and discuss whether the paternal bias observed here might be ASD-specific or due to technical artifacts.
4. They should report where the newly discovered DNMs (including PZMs and multinucleotide DNMs) locate with respect to ASD-associated genes and other de novo events (CNVs, SVs, tandem repeats). The authors note that “de novo SVs map to complex regions ill-suited for characterization by read-based alignments,” but LRS should outperform SRS for de novo SV detection, making such a comparison especially valuable in this study, even if such comparison is performed only in non-complex regions.
5. Compare the rate and mutational signature of DNMs and PZMs across different ancestries, given that the cohort includes participants of European, Asian, Indigenous American, and African ancestry. This may reveal population-specific patterns or mutational biases.
6. They should also investigate whether PZMs correlate with child age at the time of sample collection.
7. Why are there no PZMs in males (Figure 5B)?
8. The manuscript refers to the cohort as “trios”, but some families appear to be quartets (proband, unaffected sibling, and both parents). Please clarify the exact family structures and sequencing strategy (e.g., were siblings sequenced in all families?).
9. The sentence on page 4 is unclear: “all but one have a proband affected with simplex autism”. Does this mean one family

was sequenced without an autism diagnosis? Yet the text also states “a total of 157 samples from 42 families affected with simplex autism...”.

10. The abstract and results sometimes attribute findings solely to LRS, but three platforms (HiFi, ONT, Illumina) were used for discovery and validation. Reliance on three-platform validation may inflate the apparent accuracy of HiFi calls. Please discuss what the discovery rate would be using single-platform LRS alone, and compare it to short-read sequencing (SRS). Consider reporting detection confidence stratified by HiFi read depth and by validation platform. Please also clarify how many samples had ONT data available and how this affected validation rates.

11. Provide a breakdown of blood vs. LCL samples and the distribution of HiFi read depth across samples.

12. Consider presenting methylation data at or near DNMs, particularly for CpG-associated events, to provide insight into mutational mechanisms.

13. Figure and presentation clarifications:

a. In Figure 1A, mark the de novo site explicitly in the IGV snapshots.

b. In Figure 2, panels B and C appear to be switched; clarify in the legend.

c. Highlight sibling pairs consistently (currently not all are blue as described) and explain the meaning of red highlights.

d. Tandem repeat mutations are mentioned in figure legends, but per the described methods, they should have been filtered out—please clarify.

e. Comment on the single sample with an unusually high number of PZMs - does this correlate with child age or sample type?

f. In Figure 3, clarify why closely spaced mutations (1 bp apart) are not considered multinucleotide mutations, indicate why MnMs cannot be phased as germline vs. postzygotic in panel A, and define asterisks in panel B.

g. Consider reordering the figures so that their presentation follows the flow of the text; currently the narrative jumps from Figure 2 (germline vs. postzygotic comparison) to Figure 5 (sex chromosome DNMs) and then back to Figure 3 (multinucleotide mutations), which is confusing for readers.

14. On page 9, revise the phrasing to “originate from their mothers” instead of “inherited from their mothers” for de novo events on X in males.

15. Clarify whether potential CNVs were removed (“dropped haplotypes”) during the filtering of clusters of candidate de novo variants.

16. Figure S1, please clarify if the PZMs are SNVs only or both SNVs and indels.

17. Please define asterisks in Figure S5.

(Remarks on code availability)

Reviewer #2

(Remarks to the Author)

(Remarks on code availability)

The README is well documented and provides instructions to run each of the six pipelines. Although no demo data is provided, the authors have provided alternative genomes on which the code can be run to reproduce the method.

Reviewer #3

(Remarks to the Author)

This manuscript reports on the careful, long read sequencing of a set of blood samples from a set of parents and children, which are used to identify de novo mutations. The main novelty of the manuscript lies in its findings of (i) a higher mutation rate in repetitive regions of the genome that have been difficult to access by previous sequencing approaches and (ii) estimates of mutation rates for types other than single base pair substitutions (iii) a claim that the proportion of post-zygotic mutations is much higher than appreciated and shows a small but significant paternal bias.

Findings (i) and (ii) seem solid, but perhaps not that novel (in large part due to previous work by the same group), though it is useful to have more precise estimates. Claim (iii) is in my opinion not convincing, because the study design is not well suited to the question. Specifically, as the authors note but only in passing in the Discussion, by their approach, they cannot distinguish mutations that arose in the early embryogenesis of the future child from somatic mutations that arose in blood or in progenitor lineages and clonally expanded. This limitation permeates much of the study but is not dealt with in the analyses.

As one example, the authors report the surprising finding that the estimated number of post-zygotic mutations in the child increase with paternal age. They provide a speculative explanation for this finding in the Discussion (potentially at odds with Shoag et al 2025 Nat Comm, which should be cited). An alternative is that some of the post-zygotic mutations (PZMs) are actually somatic mutations in blood, that their number increases with the age of the child, and that paternal age is a proxy for the age of the child. Unless I missed something, we are not provided with the age of the children, but this explanation could be tested directly, to evaluate this possibility.

I would expect that the cohort they are analyzing would include young children, in which such somatic mosaicism would be

relatively rare (e.g., Mitchell et al. 2022 Nature). Nonetheless, it will contribute to their findings to some extent, potentially distorting most of their findings (i.e., the frequency spectrum of PZMs, their mutation rate, the sex bias etc.) These concerns could be alleviated by quantifying the extent to which blood mosaicism could contribute (beyond the hand waving comment in the Discussion) and comparing their findings to those of other groups that identified PZMs, for instance Jonsson et al. 2018 Nat Gen, Jonsson et al. 2021 Nat Gen, or ref 1 (cited).

More generally, there is frustratingly little discussion and comparison with work from other labs. As one example, they do not discuss previous papers that comment on the sex bias in mutations of PZM versus germline mutations (e.g., Lindsay et al. 2019 Nature Comm; Wu et al. 2022 eLife). A number of other examples are given in the specific comments below, but these are not exhaustive.

Finally, I was puzzled by the mutation rate that they report for the X chromosome in females, where there are two chromosomes present. How do they explain that it is $\sim 10^{-8}$, when if it were 1/4th of the male rate, it would be much lower? Could the smaller sample size used to call mutations (i.e., only females) affect their findings? That possibility could be tested by doing a similar analysis for the autosomes, to make sure the conclusions are robust.

Specific comments:

Line 47, The authors state that germline mutations arise from replication errors and the repair of double strand breaks. That view has been called into question in a number of papers (see, e.g., Gao et al. 2019 PNAS; Hahn et al. 2023 Current Biology). In particular, in females, most mutations show an age effect and therefore cannot arise from replication errors (as pointed out in ref 2, cited).

Line 50, It seems imprecise to say that most germline mutations arise in gametes, since that is just the last cell state, and germline mutations can arise at any point in the cell lineage of the gamete. More generally, these distinctions have been made in a number of papers, e.g., ref 1 cited, and it would be helpful if the authors delimited what they expect to find more carefully from the onset.

As a minor point, I think they should add the word human to the claims in the first two paragraphs of the Introduction.

Line 123: Given that a PZM should be at most at frequency 50%, how is that possible?

Line 128: Are those phasing rates significantly different?

Line 131: It seems fair to test the 1.15 to 1 ratio among PZMs with a one-tailed test, since we expect no difference or a higher rates in males (rather than the two-tailed test used). But given the somewhat unusual statistical test that they use (don't we expect the number of such mutations to be Poisson?), I am curious what they do with 0 counts and ties in the data. I would also be interested in seeing the counts in a Supplementary Figure. Most importantly, I wondered if some of the apparent bias could come from a contribution of germline mosaic mutations in the parents that are not detected in blood.

Line 348-349: PZM would not contribute much or at all to parental age effects on the number of mutations, only to the intercept, so how do I reconcile their estimate of $1.1-1.21 \times 10^{-8}$ per generation with the slopes of parental age effects estimated by, for instance ref 2, and more importantly by the lack of substantial intercept in those regressions?

Lines 362-364: seems like here they should mention the results of Palsson et al. 2025 Nature

Largely out of curiosity, since I don't expect it to affect the results: the authors mention in the Introduction that most of these families include a child with an autism diagnosis. I wondered what if any effect this ascertainment could have on their results. Presumably it could (very slightly) increase the mutation rate?

(Remarks on code availability)

Reviewer #4

(Remarks to the Author)

This study by Noyes et al leverages long-read sequencing to generate a high-confidence catalog of de novo single-nucleotide variants (DNMs) in human trios, while distinguishing germline DNMs from postzygotic mutations. The authors generated new PacBio HiFi long-read data (from blood-derived DNA), Oxford Nanopore (ONT) long-read data (from lymphoblastoid cell lines), and integrated these with existing Illumina short-read data (from blood). This design allowed them to validate each candidate variant across three independent sequencing platforms and two different tissue sources, greatly reducing technical artifacts. Using long-read phasing, they classified variants as germline when present on all reads of one haplotype and as postzygotic when present on only a subset, using allele balance across platforms to resolve ambiguous cases—yielding a highly accurate map of germline and postzygotic DNMs.

I find the results of this paper important and highly relevant to our field, and I believe that establishing a clearer connection to medical genomics would further enhance its impact (see main comments).

Main comments

1. The manuscript presents a comprehensive catalog of de novo mutations (DNMs) using long-read sequencing, but does not relate these findings to their possible functional or clinical relevance. This substantially limits the broader impact of the study. I strongly encourage the authors to assess:

- The constraint of genes harboring DNMs (e.g. LOEUF or other gene-level metrics)
 - Whether any of the DNMs are known to be deleterious or pathogenic; for missense variants, predictors such as AlphaMissense or PopEve could be applied
 - The non-coding constraint at non-coding DNMs (e.g. Gnocchi)
 - Ideally, how DNMs relate to the probands' phenotypes
- Even a preliminary analysis in this direction would considerably strengthen the manuscript.

In addition, it may be valuable to contextualize the observed DNM sites using known context-dependent mutation rates (e.g. Chen et al. 2023; Seplyarskiy et al. 2023, "Roulette" model). This would also help distinguish whether the absence of these variants from population datasets is more likely explained by low mutability or by purifying selection, which is central to their interpretation.

2. In line 626, allele balance (AB) is used via a binomial test to classify variants as postzygotic (PZMs). While AB is only applied to a subset of variants, this still raises a multiple testing concern. If, as estimated, 470 variants required AB-based classification, a significance threshold of $\alpha = 0.05$ would imply approximately 23.5 false positives by chance. It would be important to clarify how many variants remain classified as PZM after applying a multiple testing correction (e.g. FDR), or to justify why such correction is not necessary.

Minor comments

- Figure 1A: it would be helpful to clarify why a sibling was shown in the first example rather than a proband, as this may confuse readers. A color legend should be added, as IGV color codes are not intuitive to all readers.
- Reporting gnomAD or TOPMed allele frequencies for DNMs would clarify how rare these variants are. A gnomAD-based allele frequency threshold could also be incorporated as part of the filtering strategy, rather than relying solely on absence within the study cohort.
- Given that HiFi/Illumina data were generated from blood and ONT data from lymphoblastoid cell lines, discordance in AB across platforms could be used to identify potential clonal expansion in blood. This could be an informative additional analysis.
- A color legend is also missing from Figure 2B, which would improve interpretability.

Overall, I support the publication of this paper after the authors address the comments above.

(Remarks on code availability)

Version 1:

Reviewer comments:

Reviewer #1

(Remarks to the Author)

The authors have made commendable efforts to address all of the comments. Their revisions have significantly enhanced the manuscript's clarity and novelty. I have no further concerns and I support its suitability for publication.

(Remarks on code availability)

Reviewer #2

(Remarks to the Author)

(Remarks on code availability)

Reviewer #3

(Remarks to the Author)

I thank the authors for taking my suggestions seriously and appreciate their considered reply.

A few remaining concerns:

1. I think the conclusion on line 125 is overstated and the Discussion is more appropriately circumspect. In particular, I am not sure how much evidence is provided by the lack of an age effect on PZMs, if the mutations occurred early in blood development, but I found the additional lines of evidence provided in the Discussion more convincing.
2. I wonder if the authors should consider a scenario in which the mutation is germline, but there is a gene conversion event in development (as mentioned line 411), and thus the mutation ends up on three haplotypes (especially given the finding that they see more PZM in SDs).
3. In the paragraph starting on line 172, the authors may want to cite: <https://pubmed.ncbi.nlm.nih.gov/39806003/>
4. The transmission rate reported for PZM on lines 384-384 seems oddly high, as I believe that the transmission rate should be equal to the allelic balance on average (since it is carried by that proportion of cells), so much less than 60% (closer to 20%).
5. Starting line 389, the main text should discuss the lack of concordance between the mutational spectrum that they identify and others have reported, as they do in the reply to reviewers.

Possible typos:

I think line 405 should read “embryonic repair of lesions that arose in the paternal germline” (i.e., lesions not mutations).

Also line 431, what does “substitution mutations” refer to?

(Remarks on code availability)

Reviewer #4

(Remarks to the Author)

The authors have responded to my comments and implemented meaningful improvements to the article. I support acceptance for publication.

(Remarks on code availability)

Reviewer #1:

Noyes et al apply long- and short-read sequencing to identify germline and postzygotic mutations across 42 families with a child diagnosed with autism spectrum disorder (ASD). The authors find a higher rate of de novo mutations (DNMs) than previously reported, largely due to the increased sensitivity of long-read sequencing (LRS). Notably, they are able to phase DNMs using LRS, which also allows them to distinguish postzygotic mutations (PZMs) from germline mutations. The authors also investigate repetitive regions of the genome, including segmental duplications and retrotransposable elements, which are difficult to assess via short-read sequencing approaches and indeed reveal a higher rate of DNMs. Overall, this study provides valuable insight into DNMs and PZMs, and particularly highlights the utility of LRS in detecting variants in “difficult” regions of the genome.

However, several aspects of the study require clarification and further contextualization. In particular, the interpretation of PZM parental bias needs reconciliation with previous findings, the novelty relative to the authors’ own previous studies should be made explicit, figure presentation should be improved for clarity, and additional subgroup analyses may uncover biologically meaningful patterns.

We thank the reviewer for their careful review and helpful comments. We have reworked the text with a focus on improving our description and analysis of PZMs, revising figures, stressing the novelty, and searching for additional signals as specifically suggested (see below for results of new analyses).

Specific Comments:

1. The authors should explicitly specify the novelty of this work compared to the authors’ previous study (Porubsky et al., Nature 2025, ref#6), as many of the core findings (e.g., rates of DNMs in repetitive DNA, proportion of postzygotic events) are similar. Without an ASD-specific comparison (e.g., probands vs. unaffected siblings), the novelty of this manuscript is limited.

A limitation and criticism of our previous manuscript (Porubsky et al., 2025) was the focus on a single family. It has been well established that there are within and between family mutation biases (Conrad et al., Nature, 2011) as well as population differences with respect to shifts in mutational signatures. Here, we address the limitation of a single family by examining mutation biases in 42 different families of differing ethnic composition (eight children of Indigenous American, three of East Asian, two of South Asian, and two of African ancestry) and more than tripling the number of transmissions assessed by LRS. This larger sample size allows new discoveries to be made moving trends to now significant findings:

- 1) For example, we can now tease apart, for the first time, mutation rates for different classes of repetitive DNA and see evidence of not only segmental duplication (SD) enrichment but also Alu retrotransposons but surprisingly NOT LINE repeats.

- 2) We also demonstrate here that the SD mutation rate shows a clear dependence on the length and percent identity of the SD with longer SDs exhibiting significantly higher rates of mutation.
- 3) There is also the new finding that mutation rate enrichment over repeats is driven disproportionately by mutation occurring postzygotically as opposed to the germline.

Per the request to investigate DNM related to autism, we performed a detailed series of comparisons between probands and siblings searching for potential mutation signatures or features unique to one group or the other. We excluded trios (n=11) and focused on families with two children (n=31)—so-called “quad” families where there was an affected and unaffected sibling. We compared the numbers of different mutation types (A), functional annotations (B), the mutational spectrum (C), and the mutation rate across different genomic regions (D) between these matched probands and unaffected siblings (Supplementary Figure 6, below). In short, we found no significant differences between probands and their unaffected siblings arguing against overall exposure-induced increases in mutagenesis as an underlying cause in these autism families. We caution, however, that the number of families here is still relatively few and larger sample sizes are expected to reveal increases in the rate of gene-disruptive mutations without affecting the overall mutation rate (Lossifov et al., 2014).

Supplementary Figure 6: Proband and sibling comparisons

- The number of mutations observed in probands and siblings from 31 quads. Based on a linear regression accounting for paternal age at birth, we see no significant difference between counts in probands and siblings in any category (p-values: germline SNV 0.822, postzygotic SNV 0.301, insertion 0.184, deletion 0.919).
- The germline and postzygotic single-nucleotide mutation spectrum in probands and siblings. Based on Benjamini-Hochberg corrected chi-squared tests, we see no significant difference for any mutation class (p-values: A>C 0.854, A>G 0.854, A>T 0.175, C>A 0.824, C>G 0.824, C>T 0.089, CpG>CpT 0.824).
- The most severe predicted functional consequences for germline and postzygotic SNVs in probands and siblings (left), and the number of SNVs observed in repetitive regions of the genome (right).
- The germline and postzygotic mutation rates in probands and siblings in different genomic regions. Probands and siblings from the same family are joined by lines indicating which child is older. Based on Benjamini-Hochberg corrected negative

binomial regression with an offset (to account for paternal age at birth and the number of callable bases for each sample and region), we see no significant difference between probands and siblings in any region for germline SNVs (p -value >0.99 for each tested region) or postzygotic SNVs ($p>0.99$ for each tested region).

We added these proband vs. sibling analyses to the paper (Supplementary Figure 6) and made the following additions to the text:

L32 (Abstract):

Using PacBio LRS data aligned to T2T-CHM13 for discovery, we assay 2.77 Gbp of the human genome and discover an average of 95 DNMs per transmission (87.5 *de novo* single-nucleotide variants and 7.8 indels) on the autosomes and sex chromosomes, with no significant difference in mutation rate or profile between probands and their unaffected siblings.

L160:

Using the Ensembl Variant Effect Predictor²⁹ to annotate the most severe predicted consequence of each variant (Figure 2A), we find potentially deleterious DNMs and PZMs. When comparing quads with both an ASD-affected proband and sibling, we see no significant enrichment of mutations in probands (Supplementary Figure 6A), nor do we see a difference in the predicted number of deleterious DNMs or PZMs in probands (Supplementary Figure 6B), although we note that these families were selected because probands did not harbor an obvious gene-disruptive mutation based on SRS. Within the coding regions of neurodevelopmental disorder (NDD)-related genes (Supplementary Table 2; Methods), we identified one stop-gain and three missense DNMs in probands and six missense DNMs in unaffected siblings. There is no significant difference in the overall burden of NDD-related DNMs ($n=610$) between probands and siblings (chi-square test, p -value=0.8, OR=0.97). The pathogenic stop-gain DNM in *SYNGAP1* and two potential candidates in the promoter of *DDX3X* and *POGZ* were reported in Sui et al.²⁸ Only two DNMs were located within 1 kbp of a *de novo* structural variant (SV) (Supplementary Table 3, Supplementary Figure 7A,B), and both lie in intronic regions.

L220:

We see no significant differences between either the germline or postzygotic single-nucleotide substitution spectra when comparing directly between probands and their unaffected siblings (Supplementary Figure 6C).

L370:

Thus, limiting to high-confidence regions, we calculate a lower-bound autosomal germline mutation rate of 1.30×10^{-8} substitutions per base pair per generation (95% C.I. 1.27×10^{-8} - 1.34×10^{-8}) (Figure 4B) and a PZM rate of 2.30×10^{-9} substitutions per base pair per generation (95% C.I. 2.16×10^{-9} - 2.46×10^{-9}) (Figure 4C), with no significant differences between proband- or sibling-specific mutation rates (Supplementary Figure 6D).

L453 (Discussion):

In this study of 73 children from 42 families of diverse ancestry, we identified an average of 95.3 DNM events per child, with no significant differences between probands and their

unaffected siblings, arguing against overall exposure-induced increases in mutagenesis as an underlying epidemiological cause in at least these unsolved autism families.

We also added the following to the discussion, to better highlight the novelty of our study (L458):
While this is not the first study of DNMs leveraging both long- and short-read data (Noyes et al. 2022, Porubsky et al. 2025), this larger and more diverse set of samples has yielded more DNM and PZM calls than previous long-read studies. Thus, it has allowed new insights regarding *de novo* genome-wide patterns of variation, including: (1) mutation rate variation between different classes of repeats, including a significant enrichment in SINE repeat elements not observed in LINEs; (2) mutation enrichment in SDs, a feature directly correlated with higher sequence identity; (3) postzygotic enrichment in SDs and repetitive regions; and (4) MnMs that could only be validated with the availability of near-complete genome assemblies.

2. The modest paternal bias of PZMs is difficult to interpret biologically, as PZMs should occur equally on maternal and paternal alleles. Could this reflect misclassification of some germline variants as postzygotic? For indels, they commented that “a more severe paternal bias than observed for SNVs and is likely due in part to misclassified germline events”. Why does it not apply to SNVs? Consider reanalyzing PZMs restricted to those with strong SRS support (assuming SRS data have higher depth for improved classification). Discuss whether read depth influences PZM calls.

The reviewer raises a valid concern and misclassification is a possibility. We attempted to mitigate this effect by focusing on SNVs as opposed to indels. SNV positions tend to be biallelic, so it is straightforward to determine the presence or absence of a *de novo* variant on every read assigned to a parental haplotype. The presence of three haplotypes associated with a *de novo* SNV is taken as strong evidence in favor of a postzygotic event. Indels, on the other hand, are noisier in sequencing data, especially in HiFi and ONT reads. Because we often see many alleles, it can be difficult to determine whether a *de novo* variant is perfectly linked to a parental haplotype, leading to more germline events potentially misclassified as PZMs. Unfortunately, because we make our germline/postzygotic assignment by phasing reads, SRS data cannot be used to strengthen the assignment for the majority of indels, since the reads are too short to span phasing-informative SNPs. Nevertheless, we investigated the read-depth characteristics by sequencing platform and found that read depth does not have a significant influence on the number of germline or postzygotic SNVs (Supplementary Figure 4A) or indel calls (Supplementary Figure 4B).

In response, we included this analysis as Supplementary Figure 4, below, and made the following additions to the text:

L146:

There was no significant relationship between sequencing depth in any platform and the total number of germline or postzygotic SNVs observed (Supplementary Figure 4A).

L264:

Just as for SNVs, indel variants counts are robust to sequence depth (Supplementary Figure 4B).

Supplementary Figure 4: Platform read depth versus *de novo* variant counts

- A. The number of SNV calls plotted against the depth of coverage in each sequencing platform. Based on negative binomial regression, we do not see a significant relationship between read depth and germline SNV count (p-values: HiFi 0.666, ONT 0.832, Illumina 0.128) or postzygotic SNV count (p-values: HiFi 0.463, ONT 0.917, Illumina 0.447).
- B. The number of indel calls plotted against the depth of coverage in each sequencing platform. Based on negative binomial regression, we do not see a significant relationship between read depth and germline SNV count (p-values: HiFi 0.253, ONT 0.944, Illumina 0.866) or postzygotic SNV count (p-values: HiFi 0.8925, ONT 0.0815, Illumina 0.698).

3. The authors' previous study (Porubsky et al., Nature 2025, ref #6) reported that 16% of de novo single-nucleotide variants are postzygotic, with no paternal bias, including early germline mosaic events. Please reconcile this discrepancy and discuss whether the paternal bias observed here might be ASD-specific or due to technical artifacts.

In Porubsky et al., we actually found a more severe paternal bias for PZMs (1:38:1, paired two-sided Wilcoxon signed-rank test p-value = 0.091), but it did not rise to significance because the number of events was few. In the present study, the paternal bias is more modest (1.15:1, p-value = 0.0304). Based on a two-sided Wilcoxon test, we do not see a significant difference between the paternal:maternal ratio in the platinum callset and our current callset. One potential explanation for why the Porubsky callset parental ratio was not significant is that simply there wasn't enough power in the callset (which only had 119 PZMs)—similar to early reports of no maternal effect for DNM. Even this slight discrepancy argues for why a survey of more families is critical.

We examined the distribution of parental bias across all our 72 samples (Supplementary Figure 8A) and found that the mean fraction of paternal mutations was 0.53, with a standard deviation of 0.18. Unsurprisingly, the parental bias is noisy, because we are limited in our ability to identify low-AB PZMs with sequencing depth of only 30x.

To test whether the paternal bias may be in part due to technical artifacts, we applied more stringent filters to our callset, including applying multiple testing correction to our AB binomial tests, restricting our calls to only those in unique regions of the genome, eliminating all calls without at least two reads of support in HiFi and one other platform, and finally, combining all filters, reducing our callset by 66 PZMs in total. In all five callsets, the paternal:maternal ratio remains near 1.15:1, and all rise to significance by the Wilcoxon test (Supplementary Figure 8B).

Nevertheless, this large standard deviation reflects the difficulty of calling PZMs with our method and introduces enough uncertainty that we decided to remove the 1.15 claim from the abstract and leave it to the results section, flagging potential limitations.

We included this analysis as Supplementary Figure 8 and removed the following text from the abstract, L39:

with a modest but significant bias toward paternal haplotypes (1.15:1)

Finally, we made the following additions to the text:

L189:

Across samples, we observe considerable variance in the paternal bias (an average of 51% of phased PZMs are on paternal haplotypes, with a standard deviation of 18%, Supplementary Figure 8A). Even when applying more stringent filters, such as restricting

to unique regions and requiring more read support, the paternal bias remains significant at 1.15:1 (Supplementary Figure 8B).

L509:

Although this is the first report of a significant paternal bias for PZMs (1.15:1), previous studies of human PZMs have also observed some evidence of paternal bias that did not rise to significance (Porubsky 1.17:1, Sasani 1.08:1)^{1,11}. Further, Lindsay et al. described a significant paternal bias in 55 mouse PZMs that was notably absent from humans¹⁰, although they only characterized a total of 25 human PZMs, and work from Harland et al. identified a depletion on paternal haplotypes when examining 79 mosaic mutations in cattle. Because mosaic PZMs have definitionally low AB and are subject to greater sampling error than germline events, it is not surprising to see disagreement across studies with low sample sizes. Larger studies will be required to determine what, if any, effect parental haplotype has on the mutational landscape of the early embryo. Paired with a slight paternal age effect of 0.26 PZMs per year, this paternal bias could suggest a PZM mechanism involving embryonic repair of mutations that arise in the paternal germline as a father ages (Shoag et al. 2025, Kunisake et al. 2024).

Supplementary Figure 8: Paternal bias in postzygotic SNVs

- A. Paternal and maternal fractions of n=917 postzygotic SNVs across n=73 samples.
- B. Paternal and maternal fractions of CEPH 1463 (from Porubsky et al. 2024) compared to the final PZM callset with additional filters applied. Unique regions were defined as uniquely mappable with kmer size 250 by Karimzadeh et al. 2018. P-values calculated using a two-sided Wilcoxon signed-rank test.

4. They should report where the newly discovered DNMs (including PZMs and multinucleotide DNMs) locate with respect to ASD-associated genes and other de novo events (CNVs, SVs, tandem repeats). The authors note that “de novo SVs map to complex regions ill-suited for characterization by read-based alignments,” but LRS should outperform SRS for de novo SV detection, making such a comparison especially valuable in this study, even if such comparison is performed only in non-complex regions.

We performed a comprehensive annotation of all DNMs and their relationship to ~1,800 genes that have been associated with autism and developmental delay. With the exception of a stop-gain mutation in *SYNGAP1*, most of the DNMs, including those within the promoter region of

well-known autism genes (*DDX3X* and *POGZ*), are simply candidates for further testing. While this is the focus of the associated companion manuscript (Sui et al., medRxiv, 2025, accepted in principle at Nature Communications), we summarize the findings in the main text as follows:

We also include Supplementary Table 2 as requested by Reviewer #4 and include a detailed description to the Methods section:

In the main text, we added (L166):

Within the coding regions of neurodevelopmental disorder (NDD)-related genes (Supplementary Table 2; Methods), we identified one stop-gain and three missense DNMs in probands and six missense DNMs in unaffected siblings. There is no significant difference in the overall burden of NDD-related DNMs ($n=610$) between probands and siblings (chi-square test, $p\text{-value}=0.8$, $OR=0.97$). The pathogenic stop-gain DNM in *SYNGAP1* and two potential candidates in the promoter of *DDX3X* and *POGZ* were reported in Sui et al.²⁸ Only two DNMs were located within 1 kbp of a *de novo* structural variant (SV) (Supplementary Table 3, Supplementary Figure 7A,B), and both lie in intronic regions.

Methods:

Variant consequences were annotated on both T2T-CHM13v2.0 and GRCh38 references using UCSC LiftOver⁵⁹ and the Ensembl Variant Effect Predictor (VEP, v111.0 and v110.1)²⁹. DNMs overlapping with NDD genes (1,238 SFARI genes, 664 NDD genes from Fu et al.²⁶, 615 NDD genes from Wang et al.⁶⁰, and 102 ASD genes from Satterstrom et al.⁶¹) are flagged in Supplementary Table 2. We further incorporated gene-level constraint metrics (pLI and LOEUF) from gnomAD v4.1.

Variant-level functional and pathogenicity annotations, including ClinVar clinical significance, SIFT, REVEL, AlphaMissense phred, and AlphaMissense scores from dbNSFP v4.8a⁶², EVE and popEVE⁶³, and MisFit⁶⁴, were added using GRCh38 coordinates. Noncoding features were annotated with the comREG pipeline (<https://github.com/EichlerLab/asap>) described in Sui et al.²⁸, incorporating the genomic Gnocchi score⁶⁵, regulatory regions (from UCSC and the developing brain⁶⁶), and repetitive regions. Additionally, using the transmission curation pipeline described in Sui et al.²⁸, we identified 25 *de novo* SVs (including TR expansions and contractions) in the 20 probands and 16 in the 13 unaffected siblings (Supplementary Table 3). We then quantified the genomic distance between DNMs and these *de novo* SVs using BEDTools closest.

5. Compare the rate and mutational signature of DNMs and PZMs across different ancestries, given that the cohort includes participants of European, Asian, Indigenous American, and African ancestry. This may reveal population-specific patterns or mutational biases.

Using Somalier to assign ancestry (see our companion manuscript, Sui et al., medRxiv, accepted in principle at Nature Communications), we determined that 48 of our samples are of EUR descent, followed by 8 samples of Indigenous American descent, 3 East Asian descent, 2

African descent, and 2 of South Asian descent. Because the European ancestry group is so much larger than the others, we compared European to non-European samples. We compared the mutation spectrum between European and non-European samples for germline SNVs (Supplementary Figure 12C, below) and postzygotic SNVs (Supplementary Figure 12D) and found no significant differences for any substitution class by Benjamini-Hochberg corrected chi-squared tests. For both germline (Supplementary Figure 12A) and postzygotic SNVs (Supplementary Figure 12B), we did not find a significant correlation between ancestry and mutation rate.

We added the following text to the manuscript:

L223:

nor did we see a significant effect of predicted ancestry on the mutation spectrum (Supplementary Figure 12A,B)

L374:

Further, we saw no significant effect of predicted ancestry on the germline or postzygotic mutation rate (Supplementary Figure 12C,D).

Supplementary Figure 12: Relationship between ancestry and single-nucleotide mutations

- A. Germline single-nucleotide substitution spectrum using from each Somalier-defined ancestry group (AFR; African, AMR: Indigenous American, EAS: East Asian, EUR: European, SAS: South Asian). Based on a Benjamini-Hochberg corrected chi-square test, we saw no significant relationship between mutation spectrum and European and non-European samples (p-values: A>C 0.891, A>G 0.891, A>T 0.683, C>A 0.891, C>G 0.891, C>T 0.891, CpG>CpT 0.891).
- B. Postzygotic single-nucleotide substitution spectrum using the same ancestry groupings. Based on a Benjamini-Hochberg corrected chi-square test, we saw no significant

relationship between mutation spectrum and European and non-European samples (p-values: A>C 0.965, A>G 0.965, A>T 0.965, C>A 0.965, C>G 0.965, C>T 0.965, CpG>CpT 0.965).

- C. Germline single-nucleotide substitution rates for samples using the same ancestry groupings. Based on an ANCOVA comparing European and non-European samples adjusted for both paternal and maternal age, ancestry has no significant effect on mutation rate ($p=0.781$).
- D. Postzygotic single-nucleotide substitution using the same ancestry groupings. Based on the same ANCOVA analysis, ancestry has no significant effect on mutation rate ($p=0.159$).

6. They should also investigate whether PZMs correlate with child age at the time of sample collection.

We have this information for only a subset of and were able to assess this correlation in 12 of the children in this study. We see that for both germline and postzygotic SNVs, we tend to discover fewer mutations in older children, although the relationship between mutation count and age at sample collection is not significant based on linear regression for either mutational class. Although we were not able to evaluate every sample, this trend suggests that somatic mutations are not greatly contributing to PZM counts. Note that the decline in DNMs with sample age in the plot below can be attributed to paternal age at time of birth (p -value from linear regression 0.000673), as samples collected at younger ages tended to have older parents. For example, the sample collected at the youngest age had a mother in her late 40s and a father in early late 50s, whereas the samples collected at older ages had parents in their 20s and 30s. Given the young ages of samples in our dataset, we do not expect somatic mosaicism to contribute to our callset as much as we would for adult samples (Horebeek 2019; Mitchell 2022). Paired with the absence of relationship between sample age at collection and PZM count, we feel confident that we are reporting true postzygotic events.

We added the following text and Supplementary Figure 5:

L148:

Further, we do not see a significant correlation between sample age and PZM count, indicating that the impact of somatic mutations on our callset is minimal (Supplementary Figure 5).

Supplementary Figure 5: Child age at sample collection and single nucleotide substitution counts

The number of germline and postzygotic SNV calls in $n=12$ children, plotted against the sample age at time of biospecimen collection. By linear regression, we do not see a significant effect of age on the number of postzygotic ($p\text{-value}=0.060$). For germline SNVs, we incorporated paternal age into our regression model, and found no significant effect of sample age at time of collection on the number of mutations ($p\text{-value}=0.169$).

7. Why are there no PZMs in males (Figure 5B)?

There are PZMs on the male autosomes, just not the sex chromosomes. On the autosomes, males and females are treated the same, but we perform *de novo* calling in a diploid fashion on female X chromosomes and a haploid fashion on male X and Y chromosomes. Thus, we require male sex chromosomal variants to have $AB=1$ as we would expect them to be present on all reads from a given chromosome. This stringent filter is due to the high number of false positives we observed on male sex chromosomes, especially the Y. Consequently, any postzygotic variants would be removed from our callset, as we would not expect them to be present on every read derived from the X or Y chromosome.

To make this clearer, we added the following text to our Methods section (L827):

We applied an additional male-specific filter, requiring the allele balance for SNVs to be 1, as we would expect a true de novo event to be present on all reads deriving from an X or Y chromosome. This more stringent filter helped to eliminate variants resulting from mismatching between the sex chromosomes, and yielded a total of 19 SNVs on male X chromosomes. For SNV calls on the Y chromosome, we simply checked a child and the

father's sequencing data across all three platforms to ensure that the variant was not present in the father, resulting in 140 SNVs. *We applied our $AB=1$ filter and manually validated all 140 SNVs in IGV and excluded any that had low sequencing depth (fewer than 3 reads) and extensive local variation, resulting in a final callset of 24 SNVs on male Y chromosomes. Please note that as a result of our stringent AB filter, we were not able to evaluate postzygotic variants on the male sex chromosomes, as they would necessarily have $AB < 1$.*

8. The manuscript refers to the cohort as "trios", but some families appear to be quartets (proband, unaffected sibling, and both parents). Please clarify the exact family structures and sequencing strategy (e.g., were siblings sequenced in all families?).

The reviewer is correct—we analyze both trios and quartets. When performing our analyses, however, we considered each child in a trio setting when calling DNMs, but it is true that 31 families were quartet structures while 11 were trios. We simplified the manuscript title to:

Long-read sequencing of families reveals increased germline and postzygotic mutation rates in repetitive DNA

We also added a visualization to the supplement to make this clearer:

Supplementary Figure 1: Overview of Study Design

- A. Pedigrees of families, including 31 quads and 11 trios. Note that one family has a sibling but not an ASD-affected proband; this was due to lack of proband data availability at time of study.
- B. We sequenced all three or four members of trios and quads with three sequencing platforms: Oxford Nanopore (ONT) on cell line-derived DNA ($n=72/72$ children), PacBio HiFi on blood-derived DNA ($n=63/72$ children) and cell line-derived DNA ($n=25/72$ children), and Illumina on blood-derived DNA ($n=72/72$ children).

9. The sentence on page 4 is unclear: “all but one have a proband affected with simplex autism”. Does this mean one family was sequenced without an autism diagnosis? Yet the text also states “a total of 157 samples from 42 families affected with simplex autism...”.

This is correct. One of the families does have an ASD-affected proband, but the proband’s data was not available at the time analyses were performed. As both parents and the unaffected sibling were fully sequenced, we chose to include the family as a trio without the proband.

We modified that sentence to read:

These families are part of the Simons Simplex Collection (SSC, n=40) and Study of Autism Genetics Exploration (SAGE, n=2), and all but one have a proband affected with simplex autism, *as the sequencing data for that family’s proband were not available.*

10. The abstract and results sometimes attribute findings solely to LRS, but three platforms (HiFi, ONT, Illumina) were used for discovery and validation. Reliance on three-platform validation may inflate the apparent accuracy of HiFi calls. Please discuss what the discovery rate would be using single-platform LRS alone, and compare it to short-read sequencing (SRS). Consider reporting detection confidence stratified by HiFi read depth and by validation platform. Please also clarify how many samples had ONT data available and how this affected validation rates.

HiFi data were used primarily for variant discovery, while Illumina and ONT were used exclusively for validation, and every sample had ONT data available. Detailed discussion of single-platform LRS- and SRS-calling methods can be found in our previous publication, Noyes et al., 2022, which uses the same data from one of the families included in the present study. In summary, SRS data do identify a small number of DNMs that cannot be reliably called in LRS, but using LRS allows the discovery of 20% more mutations. Without filtering, SRS callers perform an accuracy of 78.9% (meaning that 78.9% of variant calls were observed in both HiFi and ONT data), while the LRS callers GATK and DeepVariant have accuracy of 80.7% and 87.6%, respectively.

We clarified the sequencing and discovery strategy with the addition of Supplementary Figure 1 and made the following additions to the text:

L32 (abstract):

Using PacBio LRS data aligned to T2T-CHM13 for discovery, we assay 2.77 Gbp of the human genome and discover an average of 95 DNMs per transmission (87.5 *de novo* single-nucleotide variants and 7.8 indels) on the autosomes and sex chromosomes, with no significant difference in mutation rate or profile between probands and their unaffected siblings.

L93:

For all 73 children, we leveraged long-read PacBio high-fidelity (HiFi) sequencing data derived from blood for variant discovery, and both long-read Oxford Nanopore

Technologies (ONT) and short-read Illumina data for validation purposes (Supplementary Figure 1B).

11. Provide a breakdown of blood vs. LCL samples and the distribution of HiFi read depth across samples.

Detailed information on sequencing data is available in Supplemental Table 1 in our companion manuscript (Sui et al., medRxiv, 2025, accepted in principle at Nature Communications). In short, we sequenced to a target coverage of 30x, using cell line data to top off samples that failed to reach that mark. In total, 9 samples have HiFi data derived exclusively from cell lines, 47 samples are exclusively from peripheral blood, and the remaining 16 samples have a mixture of blood- and cell-line-derived HiFi data (Supplementary Figure 2). These 9 samples can be identified in Figure 1B, where they are labeled in red.

We included Supplementary Figure 2 to better represent these data and added this sentence to call more attention to the 9 samples without blood-derived HiFi reads (L123):

For samples with only cell-line derived HiFi data (n=9), we required a higher read support threshold in blood-derived Illumina data.

Supplementary Figure 2: HiFi sequencing depth by data source

HiFi sequencing depth across all samples in our dataset, sequenced to a target coverage of 30X. In total, 9 samples have HiFi data derived exclusively from cell lines, 47 samples are

exclusively from peripheral blood, and the remaining 16 samples have a mixture of blood- and cell line-derived HiFi data.

12. Consider presenting methylation data at or near DNMs, particularly for CpG-associated events, to provide insight into mutational mechanisms.

This is an interesting idea. To investigate methylation changes at or near the 6,062 DNMs identified across the 73 children, we analyzed methylation levels within ± 50 bp of the 5,826 DNMs that successfully lifted over to GRCh38. Using ONT-derived 5mC profiles (Sui et al., 2025) from 24 probands and 26 siblings, we computed mean methylation in 10 bp windows around each DNM as well as the 1 bp CpG DNM site across samples. We further stratified the DNMs by CpG versus non-CpG context and assessed methylation differences between individuals carrying the DNM and those without. Among these DNMs, 1,147 (19.7%) arose at CpG sites, whereas the remainder occurred at non-CpG sites. When evaluating non-CpG sites, methylation remains consistently undetectable across samples, regardless of mutation status. Predictably, when comparing the same CpG site across samples with and without a DNM at that site, there is a drop in methylation in the mutated sample, as the methylation target is no longer present. Surprisingly, we notice a corresponding drop within the 10 bp of the 5' side of *de novo* CpG events. A possible explanation could be a density-dependence on methylation, although this phenomenon has not been previously characterized to our knowledge.

We added the following to the main text (L212):

As expected, when comparing the same CpG site across samples with and without a DNM, we see a consistent dip in methylation in the mutated sample (Supplementary Figure 11). Surprisingly, we notice a corresponding drop within the 10 bp of the 5' side of *de novo* CpG events suggesting a potential local epigenetic effect of DNM.

And a supplementary figure:

Supplementary Figure 11: Methylation abundance across ± 50 bp surrounding 5,826 DNMs. Among the 6,062 DNMs initially identified in 41 probands and 32 unaffected siblings using the T2T-CHM13 reference from HiFi; 5,826 DNMs were retained after liftover to GRCh38 for methylation analyses. Mean methylation levels in 10 bp windows centered on each DNM (with a 1 bp window at the DNM site) were quantified using ONT-derived 5mC profiles from a subset of 24 probands and 26 unaffected siblings. Methylation level at each site was computed at the diploid level as the proportion of modified reads relative to the total reads, $N_{\text{mod}} / (N_{\text{mod}} + N_{\text{other_mod}} + N_{\text{canonical}} + N_{\text{diff}} + N_{\text{nocall}})$, where counts were obtained from Modkit (v0.3.1, <https://github.com/nanoporetech/modkit>). Of the original 5,826 DNMs, 1,147 (19.7%) occurred at CpG sites, with the remainder classified as non-CpGs. Curves represent the average methylation fraction across windows, stratified by CpG vs. non-CpG context and by sample type: DNM carriers are marked with crosses, and non-carriers with dots. Shaded regions indicate 95% confidence intervals, and the dashed vertical line marks the DNM-containing window (1 bp window).

13. Figure and presentation clarifications:

a. In Figure 1A, mark the de novo site explicitly in the IGV snapshots.

We marked the site with a star, consistent with Supplementary Figure 1.

b. In Figure 2, panels B and C appear to be switched; clarify in the legend.

The panels are correctly labeled, but to make them clearer, we added a missing color legend to Figure 2B. Panels B and C of Figure 1 were indeed switched and have been relabeled to match the legend.

c. Highlight sibling pairs consistently (currently not all are blue as described) and explain the meaning of red highlights.

The blue highlight of sibling pairs is from an outdated version of the figure; we instead opted to order samples by increasing paternal age at birth. We revised the figure of the legend to read:

Samples are ordered by increasing paternal age at birth (with the father at 21.3 years of age at time of birth (at top), and the father at 52.5 years of age at time of birth (bottom)).

Samples highlighted in red (n=9) have no blood-derived HiFi read data available.

d. Tandem repeat mutations are mentioned in figure legends, but per the described methods, they should have been filtered out—please clarify.

Mentions of tandem repeats were incorrectly carried over from an outdated version of the figure. We removed them as we were not able to confidently assess tandem repeat mutations and consequently filtered them out from our final callset.

e. Comment on the single sample with an unusually high number of PZMs - does this correlate with child age or sample type?

As far as we know, the unusually high number of PZMs does not correlate with the age of the sample's parents or the sample's phenotype. This child has exclusively cell-line-derived HiFi data, but we require support of these PZM calls in blood-derived Illumina data and do not see this pattern in samples with the same data profile. Even after applying our most stringent filters to these mutations, this sample still appears to be a jackpot event.

f. In Figure 3, clarify why closely spaced mutations (1 bp apart) are not considered multinucleotide mutations, indicate why MnMs cannot be phased as germline vs. postzygotic in panel A, and define asterisks in panel B.

In Figure 3, the variants are colored by which calling method was used for discovery: in green and purple are the DNMs and PZMs from discovery method described in the "SNVs and small indels" section, whereas the gray MnM variants were specifically discovered as part of our multinucleotide process and were validated by confirming their presence in a sample's assembled genome. All MnM variants validated by genome assembly are germline in origin, as postzygotic variants are not expected to be incorporated into a diploid genome assembly, since they represent a third haplotype.

We added the following text to the legend:

- A. Distribution of distance between pairs of variants, including SNVs and indels, within 500 bp of each other. *Variants labeled as germline and postzygotic were validated individually by confirming their presence across orthogonal read data, while multinucleotide mutations (MnMs) were also validated in combination with neighboring variants by confirming their presence in a sample's genome assembly. MnMs are all germline in origin.*
- B. Mutation spectrum of individual de novo mutations (DNMs) and multinucleotide mutations (MnMs), including paired SNV calls within 500 bp of each other. *For each distance bin, Benjamini-Hochberg corrected chi-squared tests were used to evaluate the mutation spectrum relative to the spectrum of DNMs with no neighbors within 500bp. Asterisks indicate mutation types enriched in a specific bin: C>A substitutions are significantly enriched in the 1bp bin ($p=0.00156$), and C>G substitutions are significantly enriched in the 51-100bp bin ($p=0.00162$). No other enrichments relative to unclustered mutations rose to significance ($p>0.05$).*

g. Consider reordering the figures so that their presentation follows the flow of the text; currently the narrative jumps from Figure 2 (germline vs. postzygotic comparison) to Figure 5 (sex chromosome DNMs) and then back to Figure 3 (multinucleotide mutations), which is confusing for readers.

Thank you for the suggestion, we agree that the figure ordering was confusing. Instead of reordering the figures, we opted to instead reorganize the text. The "Sex Chromosome DNM" section has been moved to follow the "Mutation rate" section. As such, the discussion of sex chromosome mutation rates has been moved to the "Sex Chromosome DNM" section, so now these figures and panels should appear in the same order as they are mentioned in the text.

14. On page 9, revise the phrasing to “originate from their mothers” instead of “inherited from their mothers” for de novo events on X in males.

Thank you for the suggestion as this greatly improves clarity.

New text:

On the X chromosome, we assigned 166 and 48 germline mutations to paternal and maternal haplotypes, respectively, including 19 SNVs observed in males that must have *originated in their mothers*

15. Clarify whether potential CNVs were removed (“dropped haplotypes”) during the filtering of clusters of candidate de novo variants.

Regions where three or more SNVs were clustered within a sliding 1 kbp window of each other were removed from standard DNM calling pipeline and were instead processed using a more stringent filtering process. After establishing their presence in orthogonal sequencing reads, we validated these variants by confirming their presence in a child’s genome assembly and absence in both parental genome assemblies. Through this process, we were unable to discover any candidate CNVs. Our variant discovery and validation process was not designed with CNVs in mind; for a more careful treatment, please reference the Sui et al. companion manuscript.

16. Figure S1, please clarify if the PZMs are SNVs only or both SNVs and indels.

Figure S1 (now Supplementary Figure 3) depicts only SNVs. We updated the legend to be more specific:

Original:

- A. Allele balance (AB) of germline SNVs across all three platforms for variants with concordant and discordant AB by chi-squared shows that most variants have AB around 0.5.
- B. PZMs tend to have low AB across platforms.

New:

- A. AB of n=5,145 germline SNVs (DNMs) in PacBio HiFi, ONT, and Illumina read data. For each variant, a chi-squared test was used to determine whether its AB is concordant across platforms ($p > 0.05$). A total of 89.1% of DNMs are concordant across platforms.
- B. AB of n=917 postzygotic SNVs (PZMs) in the same sequencing read data as A. AB concordance was again defined by chi-squared test ($p > 0.05$). A total of 85.7% of DNMs are concordant across platforms.

17. Please define asterisks in Figure S5.

In Figure S5 (now Supplementary Figure 15), the asterisks represent a significant enrichment of mutations in the region relative to across the autosomes as a whole, as determined by a Welch two-sample t-test with Benjamini-Hochberg correction. The figure legend has been updated to name the test and include p-values:

The germline (left) and postzygotic (right) single nucleotide mutation rates across repetitive regions for n=73 samples. Segmental duplications (SDs) are stratified by percent identity (% ID). Asterisks indicate a region is enriched for (either germline or postzygotic) mutations relative to the autosomes as a whole, as calculated with a Welch two-sample t-test with Benjamini-Hochberg correction. P-values for germline SNVs: exons 0.73, LINEs 0.36, SINEs 0.00025, Alus 0.000011, Simple repeats 0.25, All SDs 0.045, SDs 90-95% ID 0.38, SDs 95-98% ID 0.16, SDs 98-99% ID 0.33, SDs >99% ID 0.092, Centromeres 0.016, Centromeric transition 0.73, Centromere monomeric 0.058, Centromere HSAT 0.011, acrocentric p-arms 0.0085. P-values for postzygotic SNVs: exons 0.24, LINEs 0.96, SINEs 0.24, Alus 0.064, Simple repeats 7.7e-5, All SDs 1.91e-6, SDs 90-95% ID 0.96, SDs 95-98% ID 0.048, SDs 98-99% ID 0.037, SDs >99% ID 2.9e-5, Centromeres 0.037, Centromeric transition 0.064, Centromere monomeric 0.053, Centromere HSAT 0.0019, acrocentric p-arms 7.7e-8.

Reviewer #2:

The README is well documented and provides instructions to run each of the six pipelines. Although no demo data is provided, the authors have provided alternative genomes on which the code can be run to reproduce the method.

We appreciate Reviewer #2 careful assessment of the functionality of the six computational pipelines. According to our IRB and consent, the sequence data from the autism families can only be accessed through public repositories as part of SFARI Base and NDA and, therefore, proxy genomes for this purpose were provided. All data are publicly available, although patient data are under controlled/restricted access, so the alternative genomes were preferable for demo purposes.

Reviewer #3:

This manuscript reports on the careful, long read sequencing of a set of blood samples from a set of parents and children, which are used to identify de novo mutations. The main novelty of the manuscript lies in its findings of (i) a higher mutation rate in repetitive regions of the genome that have been difficult to access by previous sequencing approaches and (ii) estimates of mutation rates for types other than single base pair substitutions (iii) a claim that the proportion of post-zygotic mutations is much higher than appreciated and shows a small but significant paternal bias.

Findings (i) and (ii) seem solid, but perhaps not that novel (in large part due to previous work by the same group), though it is useful to have more precise estimates. Claim (iii) is in my opinion not convincing, because the study design is not well suited to the question. Specifically, as the authors note but only in passing in the Discussion, by their approach, they cannot distinguish mutations that arose in the early embryogenesis of the future child from somatic mutations that arose in blood or in progenitor lineages and clonally expanded. This limitation permeates much of the study but is not dealt with in the analyses.

As one example, the authors report the surprising finding that the estimated number of post-zygotic mutations in the child increase with paternal age. They provide a speculative explanation for this finding in the Discussion (potentially at odds with Shoag et al 2025 Nat Comm, which should be cited). An alternative is that some of the post-zygotic mutations (PZMs) are actually somatic mutations in blood, that their number increases with the age of the child, and that paternal age is a proxy for the age of the child. Unless I missed something, we are not provided with the age of the children, but this explanation could be tested directly, to evaluate this possibility.

The reviewer is correct that our study was not designed to distinguish PZMs from somatic mutations in the blood that have risen to high frequency. Our previous analysis, using the same discovery and validation strategy with a multigenerational pedigree, indicated that over half of called PZMs must have been sufficiently early to be present in the germline because we observed evidence of transmission to the next generation.

Because the paternal bias for PZM concern was noted by two of the reviewers, we decided to temper the third claim by removing this point from the abstract. We also performed additional analyses to mitigate potential misclassification, including applying additional filters, such as more stringent read support and restricting our callset to unique regions of the genome. Regardless of filters applied, the paternal bias for PZMs did remain significant (see figure below), but the reviewer is correct that we cannot distinguish early postzygotic versus later somatic mutations.

Paternal and maternal fractions of CEPH 1463 (from Porubsky et al., 2024) compared to the final PZM callset with additional filters applied. Unique regions were defined as uniquely mappable with k-mer size 250 by Karimzadeh et al., 2018. P-values calculated using a two-sided Wilcoxon signed-rank test.

In addition, we examined the effect of the age of the child at time of biospecimen collection, for the 12 children for whom we had this data (see Supplementary Figure 5 below). We found no significant effect of child age on the number of PZMs called. Note that the decline in DNMs with sample age in the plot below can be attributed to paternal age at time of birth (p-value from linear regression 0.000673), as samples collected at younger ages tended to have older parents. For example, the sample collected at the youngest age had a mother in her late 40s and a father in early late 50s, whereas the samples collected at older ages had parents in their 20s and 30s.

Supplementary Figure 5: Child age at sample collection and single-nucleotide substitution counts

The number of germline and postzygotic SNV calls in $n=12$ children, plotted against the sample age at time of biospecimen collection. By linear regression, we do not see a significant effect of age on the number of postzygotic SNVs ($p=0.060$). For germline SNVs, we incorporated paternal age into our regression model and found no significant effect of sample age at time of collection on the number of mutations ($p=0.169$).

Finally, although the subtle paternal age effect is intriguing, we believe there are additional novel findings here that we have made an effort to elaborate upon in the revised manuscript. For example, we can now tease apart for the first time the mutation rate for different classes of repetitive DNA for evidence of not only segmental duplication (SD) enrichment but also Alu retrotransposons yet surprisingly NOT LINE repeats. We also show that SD mutation rate shows a clear dependence on the length and percent identity of the SD with longer SDs showing significantly higher rates of mutation. There is the new finding that this mutation rate enrichment over repeats is driven disproportionately by mutation occurring postzygotically as opposed to the germline

We added the following text to the discussion to contextualize our findings with Shoag et al., 2025 (L519):

Paired with a slight paternal age effect of 0.26 PZMs per year, this paternal bias could suggest a PZM mechanism involving embryonic repair of mutations that arise in the paternal germline as a father ages (Shoag et al. 2025, Kunisaki et al. 2024).

In addition, we added the following text to the discussion to better highlight the novelty of our study (L455):

In this study of 73 children from 42 families of diverse ancestry, we identified an average of 95.3 DNM events per child, with no significant differences between probands and their unaffected siblings, arguing against overall exposure-induced increases in mutagenesis

as an underlying epidemiological cause in at least these unsolved autism families. While this is not the first study of DNMs leveraging both long- and short-read data (Noyes et al. 2022, Porubsky et al. 2025), this larger and more diverse set of samples has yielded more DNM and PZM calls than previous long-read studies. Thus, it has allowed new insights regarding *de novo* genome-wide patterns of variation, including: (1) mutation rate variation between different classes of repeats, including a significant enrichment in SINE repeat elements not observed in LINEs; (2) mutation enrichment in SDs, a feature directly correlated with higher sequence identity; (3) postzygotic enrichment in SDs and repetitive regions; and (4) MnMs that could only be validated with the availability of near-complete genome assemblies.

I would expect that the cohort they are analyzing would include young children, in which such somatic mosaicism would be relatively rare (e.g., Mitchell et al. 2022 Nature). Nonetheless, it will contribute to their findings to some extent, potentially distorting most of their findings (i.e., the frequency spectrum of PZMs, their mutation rate, the sex bias etc.) These concerns could be alleviated by quantifying the extent to which blood mosaicism could contribute (beyond the hand waving comment in the Discussion) and comparing their findings to those of other groups that identified PZMs, for instance Jonsson et al. 2018 Nat Gen, Jonsson et al. 2021 Nat Gen, or ref 1 (cited).

This is a reasonable concern, as our study was not originally designed for PZM discovery and, thus, we lack data from a second tissue to distinguish PZMs from somatic variants.

Although we only have parental age information for 12 of 73 total samples in our dataset, we know that all samples in the SSC were collected between the ages of 4 and 18 (we have two samples collected under the age of 4 that were originally sequenced as part of a different cohort). For the 12 samples with age-at-collection data, we do not see an increase in PZM count in older samples—an indication that somatic mutations are not the only source of variation in our PZM dataset.

Despite the observed accumulation of 17 mutations per year after birth in hematopoietic stem cells (Mitchell et al., 2022), most mutations will not clonally expand, especially in such young samples (Laurie et al., 2012). For mutations that did not undergo expansion, we anticipate their AB to be below our threshold for detection, as the lowest observed AB for a PZM in our callset is 0.074 and we only sequenced to 30x coverage. While it is known that clonal expansions observed later in life often begin with mutations that occur early in childhood, there are few studies of healthy children that quantify the extent of detectable expansions. Even in younger adults, these expansions can be hard to detect (Mitchell et al., 2022; Bernstein et al., 2024; Fabre et al., 2024), and given that 99% of our PZMs have $AB > 0.1$ (and are thus present in 20% of blood cells) and none are known driver mutations, the likelihood of seeing a significant contribution of clonal expansion events is low.

As stated in our Discussion, we performed variant discovery using the same technique as an earlier study (Porubsky et al., 2024), in which we were able to use transmission to a third

generation to show that approximately two-thirds of PZM calls can be found in the blood and germline. While we do not have the data to repeat that analysis for these samples, paired with the young age of our samples and the absence of an age-at-collection-time signal, we feel confident that a majority of PZMs are truly early developmental events.

Although we don't have a second tissue from these samples to compare against, we attempted to quantify AB skewing in blood and cell-line data for 15 samples with at least 10X HiFi coverage derived from both sources. We examined 1,137 variants from these samples and restricted our analysis to reads with mapq>59 to remove any false mosaic events resulting from read multi-mapping. Based on Fisher's exact tests with Benjamini-Hochberg correction, we found only one variant with significantly different AB between blood- and cell-line-derived HiFi data (Supplementary Figure 3C,D). We did not observe a tight correlation between blood and cell-line data, but we also do not observe a systematic skew in either data type (Supplementary Figure 3E,F).

When comparing our results to other studies of PZMs, it is worth noting that we have a larger PZM callset (n=917), in part because we are able to leverage long-read data to phase and discover mutations that are not perfectly linked to a parental haplotype. For example, Jonsson et al. reported phasing numbers for 582 PZMs (with 46% on paternal haplotypes) and Sasani et al. reported phasing numbers for 475 PZMs (with 52% on paternal haplotypes). When applying an identical PZM analysis to CEPH 1463 (which was also evaluated in Sasani et al.), we found that 54% of 120 PZMs were on paternal haplotypes. Compared to the current study, where 53.5% of our phased PZMs fall on paternal haplotypes, the bias is well within the range of previous analyses, and when comparing with a Wilcoxon signed-rank test, we find no significant difference between the bias reported in our callset and that from Porubsky et al. (p=0.406) or Sasani et al. (p=0.177). We did not compare directly to Jonsson et al. due to data availability.

In addition, all of these four PZM callsets have different mutational profiles: Jonsson et al. see an enrichment of postzygotic A>C, C>A, and C>T (non-CpG) mutations, while Sasani et al. report only an A>T enrichment, which is notably absent from the Jonsson paper. In Porubsky et al., we reported a postzygotic enrichment of A>T mutations and a depletion of CpG>TpG events, both of which are observed in our current callset, as well as an enrichment of A>C mutations, similar to what Jonsson et al. observed. Although there is not an established consensus on the PZM spectrum, we do replicate key findings of previous PZM studies.

We added the following text:

L485:

This enrichment may be in part the result of somatic mutations that arose and clonally expanded in blood; such mutations cannot be filtered without access to another tissue for sequencing, a clear shortcoming of our ability to generate a highly specific PZM callset. However, blood data were collected when our samples were under the age of 18, so we do not expect to see as much somatic contamination as we would in an adult dataset (Mitchel et al. 2022, Horebeek et al. 2019, Fabre et al. 2024), and there is no correlation between sample age and total PZM count (Supplementary Figure 5). Further,

over 98% of our PZMs have HiFi AB greater than 0.1, indicating their presence in at least 20% of sequenced cells—at this threshold, detection of somatic variants, even those that may have clonally expanded during early childhood, is unlikely (Bernstein et al. 2024).

L510:

Although this is the first report of a significant paternal bias for PZMs (1.15:1), previous studies of human PZMs have also observed some evidence of paternal bias that did not rise to significance (Porubsky 1.17:1, Sasani 1.08:1)^{1,11}. Further, Lindsay et al. described a significant paternal bias in 55 mouse PZMs that was notably absent from humans¹⁰, although they only characterized a total of 25 human PZMs, and work from Harland et al. identified a depletion on paternal haplotypes when examining 79 mosaic mutations in cattle. Because mosaic PZMs have definitionally low AB and are subject to greater sampling error than germline events, it is not surprising to see disagreement across studies with low sample sizes. Larger studies will be required to determine what, if any, effect parental haplotype has on the mutational landscape of the early embryo. Paired with a slight paternal age effect of 0.26 PZMs per year, this paternal bias could suggest a PZM mechanism involving embryonic repair of mutations that arise in the paternal germline as a father ages (Shoag et al. 2025, Kunisake et al. 2024).

L506:

In addition, PZMs are enriched for A>C and A>T substitutions, as previously described (Jonsson et al. 2021, Sasani et al. 2019).

We also added the following panels to Supplementary Figure 3:

Supplementary Figure 3: Allele balance (AB) concordance scatterplots

- A. AB of $n=5,145$ germline SNVs (DNMs) in PacBio HiFi, ONT, and Illumina read data. For each variant, a chi-squared test was used to determine whether its AB is concordant across platforms ($p>0.05$). A total of 89.1% of DNMs are concordant across platforms.
- B. AB of $n=917$ postzygotic SNVs (PZMs) in the same sequencing read data as A. AB concordance was again defined by chi-squared test ($p>0.05$). A total of 85.7% of DNMs are concordant across platforms.
- C. AB in cell-line- and blood-derived HiFi read data (restricted to reads with $\text{mapq}>59$) for $n=979$ germline SNVs from 15 samples. Although we do not see the expected $x=y$ relationship between reads from both data sources, there is no evidence of systematic over- or underestimation between sources. Each variant was tested for concordant AB using a Fisher's exact test with Benjamini-Hochberg correction; none were significantly different ($p\text{-value}>0.05$).
- D. AB in cell-line- and blood-derived HiFi read data (restricted to reads with $\text{mapq}>59$) for $n=66$ postzygotic SNVs from 15 samples. Similar to SNVs, we do not see a clear linear relationship between AB from data sources, but we also do not observe systematic bias. Each variant was tested for concordant AB using a Fisher's exact test with Benjamini-Hochberg correction; highlighted in yellow and circled in red is the single variant with significantly different AB across sources ($p=0.0193$).
- E. Bland-Altman plot showing the mean AB across data sources plotted against the difference between data sources for the same germline SNVs in (A). The mean difference in AB is 0.006, with a standard deviation of 0.188.

- F. Bland-Altman plot showing the mean AB across data sources plotted against the difference between data sources for the same postzygotic SNVs in (B). The mean difference in AB is 0.022, with a standard deviation of 0.197.

More generally, there is frustratingly little discussion and comparison with work from other labs. As one example, they do not discuss previous papers that comment on the sex bias in mutations of PZM versus germline mutations (e.g., Lindsay et al. 2019 Nature Comm; Wu et al. 2022 eLife). A number of other examples are given in the specific comments below, but these are not exhaustive.

We apologize for this oversight and have added these and other references to create a more balanced and integrated manuscript. A subset of these additional citations include:

L488:

However, blood data were collected when our samples were under the age of 18, so we do not expect to see as much somatic contamination as we would in an adult dataset (Mitchel et al. 2022, Horebeek et al. 2019, Fabre et al. 2024), and there is no correlation between sample age and total PZM count (Supplementary Figure 5). Further, over 98% of our PZMs have HiFi AB greater than 0.1, indicating their presence in at least 20% of sequenced cells—at this threshold, detection of somatic variants, even those that may have clonally expanded during early childhood, is unlikely (Bernstein et al. 2024).

L519:

Paired with a slight paternal age effect of 0.26 PZMs per year, this paternal bias could suggest a PZM mechanism involving embryonic repair of mutations that arise in the paternal germline as a father ages (Shoag et al. 2025, Kunisaki et al. 2024).

L554:

The fact that DNMs in SDs are significantly depleted in CpG substitutions and show an excess of transversions once again points to a role for IGC in driving some of this acceleration (Vollger et al. 2023, Palsson et al. 2025).

Finally, I was puzzled by the mutation rate that they report for the X chromosome in females, where there are two chromosomes present. How do they explain that it is $\sim 10^{-8}$, when if it were 1/4th of the male rate, it would be much lower? Could the smaller sample size used to call mutations (i.e., only females) affect their findings? That possibility could be tested by doing a similar analysis for the autosomes, to make sure the conclusions are robust.

Thank you for catching this. When we look at mutation counts, we see 48 and 166 mutations on maternally and paternally inherited X chromosomes, respectively, which is a 3.46 ratio. Based on a two-proportion Z test (two-sided), this ratio is consistent with the 3.98:1 paternal:maternal DNM ratio we observe genome-wide (p-value = 0.483).

The discrepancy between the mutation rate ratio and the DNM count ratio is in fact due to missing data—samples with 0 *de novo* SNVs on the sex chromosome were erroneously omitted from the final mutation rate calculation. Incorporating them back into the dataset, we calculate the maternal X chromosome mutation rate to be 4.55×10^{-9} , the paternal X chromosome

mutation rate to be 2.47×10^{-8} , and the Y chromosome mutation rate to be 5.14×10^{-8} . We updated both the text and Figure 5 to reflect this correction.

These mutation rates are calculated without accounting for the sex of the child, but rates do somewhat differ between male and female children, because on average we are able to assess variation in an additional 200 kbp of the female X chromosome compared to the male X. This results in a small denominator for the male mutation rate calculation, driving it up somewhat. We calculate the maternal X chromosome mutation rate in females, where the amount of callable space is higher, to be 4.33×10^{-9} , compared to 4.93×10^{-9} in males, where the amount of callable space is lower.

Original (L276-283):

With respect to the sex chromosomes, we estimate the X chromosome mutation rate to be 1.01×10^{-8} substitutions per base pair per generation in the maternal germline and 2.59×10^{-8} substitutions per base pair per generation in the paternal germline (Figure 5C). This enrichment of mutations in the paternal germline is even more stark on the Y chromosome, where we see a mutation rate of 1.16×10^{-7} substitutions per base pair per generation. Combined with the germline mutations we observed on the autosomes, we calculate the whole-genome mutation rate in the maternal and paternal germline to be 0.52×10^{-8} and 2.09×10^{-8} substitutions per base pair per generation, respectively.

New (L434):

We estimate the X chromosome mutation rate to be 0.46×10^{-8} substitutions per base pair per generation in the maternal germline and 2.47×10^{-8} substitutions per base pair per generation in the paternal germline (Figure 5C). This enrichment of mutations in the paternal germline is even more stark on the Y chromosome, where we see a mutation rate of 5.14×10^{-8} substitutions per base pair per generation, although given our ascertainment against repeat-rich regions this must be regarded as a lower bound. Combined with the germline mutations we observed on the autosomes, we calculate the whole-genome mutation rate in the maternal and paternal germline to be 0.52×10^{-8} and 2.05×10^{-8} substitutions per base pair per generation, respectively.

Revised Figure 5:

Specific comments:

Line 47, The authors state that germline mutations arise from replication errors and the repair of double strand breaks. That view has been called into question in a number of papers (see, e.g., Gao et al. 2019 PNAS; Hahn et al. 2023 Current Biology). In particular, in females, most mutations show an age effect and therefore cannot arise from replication errors (as pointed out in ref 2, cited).

We modified that introductory text to reflect that nuance:

They arise from mutational processes in the parental germline, such as double-stranded break repair or errors in DNA replication¹⁻⁵, *although the relative contribution of replication errors has been recently called into question (Gao et al. 2019, de Manuel et al. 2022, Hahn et al. 2023).*

Line 50, It seems imprecise to say that most germline mutations arise in gametes, since that is just the last cell state, and germline mutations can arise at any point in the cell lineage of the gamete. More generally, these distinctions have been made in a number of papers, e.g., ref 1 cited, and it would be helpful if the authors delimited what they expect to find more carefully from the onset.

We revised the text to be more precise:

These mutations have been called early embryonic (Ju et al., 2017, Lindsay et al., 2019) or gonosomal (Sasani et al., 2019), but they are most commonly referred to as postzygotic mutations (PZMs) to distinguish them from germline DNMs arising during the process of parental gametogenesis (Jonsson et al. 2021, Huang et al. 2014, Acuna-Hidalgo 2015).

As a minor point, I think they should add the word human to the claims in the first two paragraphs of the Introduction.

Great point! We modified the text to specify that we are referring to human-specific studies.

Line 123: Given that a PZM should be at most at frequency 50%, how is that possible?

This line is referring to germline events, not PZMs, that have AB=1. Regardless, for these 11 events, seeing AB=1 across platforms doesn't make much biological sense. For five of these DNMs, this elevated AB is an artifact of the way we count supporting reads. These events have HiFi data with the reference allele, but those reads fail to pass our base quality or mapping quality thresholds and are not included in the final AB calculation.

The remaining six variants all fall in repetitive regions, with read depth between 15 and 23, which is lower than the genome-wide average. It is likely that reads with the reference allele are

being assigned to multiple locations and are thus being excluded by our secondary alignment read filters.

We do not expect $AB=1$ to be the biological truth for any of these variants, but rather it is an indication of how challenging it can be to discover variation in these messy parts of the genome.

Line 128: Are those phasing rates significantly different?

Yes, they are significantly different by two-sample test for equality of proportions ($p=0.000119$).

We updated the text to read:

We were able to assign 98.0% and 96.1% of germline and postzygotic SNVs, respectively, to parental haplotypes, a small but significant difference (two-sample test for equality of proportions, $p=0.000119$).

Line 131: It seems fair to test the 1.15 to 1 ratio among PZMs with a one-tailed test, since we expect no difference or a higher rates in males (rather than the two-tailed test used). But given the somewhat unusual statistical test that they use (don't we expect the number of such mutations to be Poisson?), I am curious **what they do with 0 counts and ties in the data. I would also be interested in seeing the counts in a Supplementary Figure.** Most importantly, I wondered if some of the apparent bias could come from a contribution of germline mosaic mutations in the parents that are not detected in blood.

We performed a Wilcoxon test in R using the following command:

```
wilcox.test(maternal_count, paternal_count, paired = TRUE, alternative = "two.sided")
```

Ties in the data ($n=8$) are dropped, while samples with 0 PZMs on one parental haplotype ($n=2$) are retained. Since the Wilcoxon test is not the most standard, we also applied a binomial test to our PZM count data and found that the paternal bias still remains significant, with p -value=0.0366. We also fit a binomial generalized linear model to our count data, and the paternal bias is once again significant, with p -value=0.0339.

We added Supplementary Table 4 with the number of paternal and maternal PZMs and plotted the phased fractions of PZMs in Supplementary Figure 8A.

Finally, as a mosaic germline mutation in a parent would be present in the fertilized zygote, we would not expect to see the incomplete haplotype linkage pattern characteristic of our PZMs. Instead, we do find mutations in this category when we look at shared DNM calls between a proband and their sibling, so we removed all such calls from our final callset.

Supplementary Figure 8: Paternal bias in postzygotic SNVs

A. Paternal and maternal fractions of n=917 postzygotic SNVs across n=73 samples.

Line 348-349: PZM would not contribute much or at all to parental age effects on the number of mutations, only to the intercept, **so how do I reconcile their estimate of $1.1-1.21 \times 10^{-8}$ per generation with the slopes of parental age effects** estimated by, for instance ref 2, and more importantly by the lack of substantial intercept in those regressions?

This is a fair point: if we assume that some percentage of DNMs in previous studies are actually misclassified PZMs, then when plotting the parental age effect on total DNM count, we should expect to see a y-intercept that reflects some amount of standing postzygotic variation. So, when we make the claim that the true germline rate in previous studies might actually be lower than reported due to postzygotic contamination, why doesn't that contamination show up as a y-intercept in parental age plots?

It is worth noting that even in our regressions just examining the effects of paternal and maternal age on PZM count, we do not see a y-intercept significantly different from 0 ($p=0.607$, $p=0.878$ in paternal and maternal data, respectively). One potential explanation is that PZM count is quite noisy and variable between samples. In our data here, we see an outlier with over 60 PZMs, and samples with as few as two or three PZMs, so we would not expect to see a consistent y-intercept across samples.

Based on the fact that PZMs make up approximately 15% of our dataset, and that 30% of our PZMs have an $AB > 0.30$ (a standard cutoff in many but not all DNM studies), we estimated that PZM contamination might account for 5-10% of previously reported DNMs and adjusted the mutation rates down accordingly. As this is an imprecise estimate, we rephrased the text to be more conservative as follows:

Original:

Adjusting for PZMs, the germline mutation rates from previous studies would more likely fall into the range of $1.1-1.21 \times 10^{-8}$.

New:

Given that nearly 5% of our total callset is composed of PZMs with $AB > 0.30$, a standard filtering threshold in DNM studies, the true germline mutation rates in these previous studies could be overestimated by as much as 5%.

Lines 362-364: seems like here they should mention the results of Palsson et al. 2025 Nature

We modified the text to read:

The fact that DNMs in SDs are significantly depleted in CpG substitutions and show an excess of transversions once again points to a role for IGC or potentially allelic gene conversion events in driving some of this acceleration (Vollger et al. 2023, Palsson et al. 2025).

Largely out of curiosity, since I don't expect it to affect the results: the authors mention in the Introduction that most of these families include a child with an autism diagnosis. I wondered what if any effect this ascertainment could have on their results. Presumably it could (very slightly) increase the mutation rate?

Affected status seems to have very little if any effect. To test this, we excluded trios ($n=11$) and focused on families with an affected and unaffected sibling ($n=31$). As outlined in Supplementary Figure 6, we found no significant differences between probands and siblings across the numbers of different mutation types (A), functional annotations of mutations (B), the mutational spectrum (C), or the mutation rate across different genomic regions (D) between these matched probands and unaffected siblings. It appears that it is not the number of mutations that determines affected status, but rather the location of those mutations, epigenetic effects, or other factors as of yet uncovered.

We added the following analysis to the supplement:

Supplementary Figure 6: Proband and sibling comparisons

- The number of mutations observed in probands and siblings from 31 quads. Based on a linear regression accounting for paternal age at birth, we see no significant difference between counts in probands and siblings in any category (p-values: germline SNV 0.822, postzygotic SNV 0.301, insertion 0.184, deletion 0.919).
- The germline and postzygotic single-nucleotide mutation spectrum in probands and siblings. Based on Benjamini-Hochberg corrected chi-squared tests, we see no significant difference for any mutation class (p-values: A>C 0.854, A>G 0.854, A>T 0.175, C>A 0.824, C>G 0.824, C>T 0.089, CpG>CpT 0.824).
- The most severe predicted functional consequences for germline and postzygotic SNVs in probands and siblings (left), and the number of SNVs observed in repetitive regions of the genome (right).
- The germline and postzygotic mutation rates in probands and siblings in different genomic regions. Probands and siblings from the same family are joined by lines

indicating which child is older. Based on Benjamini-Hochberg corrected negative binomial regression with an offset (to account for paternal age at birth and the number of callable bases for each sample and region), we see no significant difference between probands and siblings in any region for germline SNVs (p -value >0.99 for each tested region) or postzygotic SNVS ($p>0.99$ for each tested region). Supplementary Figure 3A,B). We find that roughly 89% of DNMs and 85% of PZMs are concordant across all three sequencing platforms.

Reviewer #4:

This study by Noyes et al leverages long-read sequencing to generate a high-confidence catalog of de novo single-nucleotide variants (DNMs) in human trios, while distinguishing germline DNMs from postzygotic mutations. The authors generated new PacBio HiFi long-read data (from blood-derived DNA), Oxford Nanopore (ONT) long-read data (from lymphoblastoid cell lines), and integrated these with existing Illumina short-read data (from blood). This design allowed them to validate each candidate variant across three independent sequencing platforms and two different tissue sources, greatly reducing technical artifacts. Using long-read phasing, they classified variants as germline when present on all reads of one haplotype and as postzygotic when present on only a subset, using allele balance across platforms to resolve ambiguous cases—yielding a highly accurate map of germline and postzygotic DNMs.

I find the results of this **paper important and highly relevant to our field**, and I believe that establishing a clearer connection to medical genomics would further enhance its impact (see main comments).

We thank the reviewer for their review and helpful comments. While the companion manuscript by Sui et al., 2025, more rigorously assesses the potential functional consequences of the DNMs, we have performed additional analyses requested by this reviewer, as well as Reviewer #1, and include observations of medical significance to increase the appeal of the manuscript. We find, for example, no significant difference in the DNM rate between probands and unaffected siblings after regressing out paternal age.

Main comments

1. The manuscript presents a comprehensive catalog of de novo mutations (DNMs) using long-read sequencing, but does not relate these findings to their possible functional or clinical relevance. This substantially limits the broader impact of the study. I strongly encourage the authors to assess:

- The **constraint of genes harboring DNMs** (e.g. LOEUF or other gene-level metrics)
- Whether any of the **DNMs are known to be deleterious or pathogenic**; for missense variants, predictors such as AlphaMissense or PopEve could be applied
- The **non-coding constraint at non-coding DNMs** (e.g. Gnocchi)
- Ideally, how DNMs relate to the probands' phenotypes

Even a preliminary analysis in this direction would considerably strengthen the manuscript.

We agree with the reviewer and performed a comprehensive annotation of DNMs in Supplementary Table 3, also requested by Reviewer #1.

The following description was added to the Methods section:

Functional annotation of DNM. Variant consequences were annotated on both T2TCHM13v2.0 and GRCh38 references using UCSC LiftOver (Hinrichs et al., 2006) and the Ensembl Variant Effect Predictor (VEP, v111.0 and v110.1; McLaren et al.,

2016). DNMs overlapping with neurodevelopmental disorders (NDD) genes (1,238 SFARI genes, 664 NDD genes from Fu et al., 615 NDD genes from Wang et al., and 102 ASD genes from Satterstrom et al.) were flagged in the Supplementary Table 3. We further incorporated gene-level constraint metrics (pLI and LOEUF) from gnomAD v4.1. Variant-level functional and pathogenicity annotations, including ClinVar clinical significance, SIFT, REVEL, AlphaMissense phred, and AlphaMissense scores from dbNSFP v4.8a (Liu et al., 2020), EVE and popEVE (Orenbuch et al., NG, 2025), and MisFit (Zhao et al., NC, 2025), were added using GRCh38 coordinates. Noncoding features were annotated with the comREG pipeline (<https://github.com/EichlerLab/asap>) described in Sui et al., incorporating the genomic Gnocchi score (Chen et al. Nature, 2024), regulatory regions (from UCSC and the developing brain (Henick et al. 2025)), and repetitive regions. Additionally, using the transmission curation pipeline described in Sui et al., we identified 25 *de novo* SVs (including TR expansions and contractions) in the 20 probands and 16 in the 13 unaffected siblings (Supplementary Table 4). We then quantified the genomic distance between DNMs and these *de novo* SVs using *bedtools closest*.

Supplementary Figure 7: *de novo* SNVs near *de novo* SVs

- A. IGV screenshot of HiFi data aligned to GRCh38 for the members of 12651 - a *de novo* T>C mutation can be seen immediately to the left of the SV. Note that this mutation appears to be germline in HiFi data, but is imperfectly linked to surrounding SNPs in ONT data, and was therefore deemed to be postzygotic in origin.

- B. IGV screenshot of HiFi data aligned to GRCh38 for the members of 13610 - a *de novo* C>T mutation can be seen immediately to the left of the SV.

In addition to Supplementary Figure 7, we added the following to the main text (L158):

Within the coding regions of NDD-related genes (Supplementary Table 3; Methods), we identified 1 stop-gain and 3 missense DNMs in probands and 6 missense DNMs in unaffected siblings. There is no significant difference in the overall burden of NDD-related DNMs (n=610) between probands and siblings (Chi-square test, p-value = 0.8, OR=0.97). The pathogenic stop-gain DNM in *SYNGAP1* and two potential candidates in the promoter of *DDX3X* and *POGZ* were reported in Sui et al. Only two DNMs were located within 1 kb of a *de novo* SV (Supplementary Table 4, Supplementary Figure 7), and both lie in intronic regions.

In addition, it may be valuable to contextualize the observed DNM sites using known context-dependent mutation rates (e.g. Chen et al. 2023; Seplyarskiy et al. 2023, “Roulette” model). This would also help distinguish whether the absence of these variants from population datasets is more likely explained by low mutability or by purifying selection, which is central to their interpretation.

We investigated the Roulette model against our DNM sites, specifically testing whether we observed elevations in TFBS. We followed the instructions outlined on their GitHub page to rescale the predicted mutation rates for our own *de novo* callset, using gnomAD v2.1.1 synonymous variants as background sites. For variants that we could lift to GRCh38 (n=4,980 DNMs, n=838 PZMs), we annotated both the scaled and unscaled mutation rates and examined which variants had an elevated predicted rate. Interestingly, approximately half of variants were in annotated TFBS. When we compare germline maternal and paternal SNVs, we see a significant paternal enrichment in TFBS (maternal fraction 0.46, paternal fraction 0.51, two-sample test for equality of proportions p-value=0.013), which is in line with the elevated TFBS mutation rate observed in testis by Seplyarskiy et al.

When looking at variants with high predicted mutation rates ($>1e-5$, n=890 DNMs, n=117 PZMs), we find that all but six are CpG mutations, with 50.6% falling in TFBS. Of the remaining six variants, one is in a TFBS, one is in an SD, two are in repeat elements (hat-Charlie and LINE), one is in an intron, and the final is in a region with no repeats but observed copy number variation in SGDP.

Supplementary Figure 10: Roulette-predicted mutation rates and TFBS

Roulette predicted mutation rates for germline SNVs (left) and postzygotic SNVs (right) successfully lifted to GRCh38. Rates were scaled to our *de novo* callset, using gnomAD v2.1.1 synonymous variants as background sites. Variants are colored by presence/absence in transcription factor binding sites (TFBS). All but 6 mutations with rate $> 1e-5$ are at CpG sites.

It is worth noting that 30% of DNMs and 26.5% of PZMs reported here are in fact observed from gnomAD. Of these 1,525 events, 820 (99.2% CpG) have predicted high mutability by Roulette, and an additional 49 are in CpG sites that are not predicted to be highly mutable. Conversely, 70 (100% CpG) of the 3,455 SNVs absent from gnomAD are in predicted highly mutable regions, and only 140 CpG mutations were not previously observed in gnomAD. Paired with low functional constraint LEOUF scores, this suggests that most of our mutations are absent from gnomAD due to low mutability.

However, in regions where we observed the highest mutation rates, like SDs, many variants could not be lifted to GRCh38-space, and we have previously shown that these variants are not detectable with short-read data employed by population-level studies (Noyes et al., 2022). For these regions, we cannot infer that low predicted mutability or absence in gnomAD is a true indication of the mutation rate.

We included Supplementary Figure 10 and the following text (L184):

When we compare germline maternal and paternal SNVs, we see a significant paternal enrichment in TFBS (maternal fraction 0.46, paternal fraction 0.51, two-sample test for

equality of proportions p -value=0.013), which is consistent with the elevated TFBS mutation rate observed in testis by Seplyarskiy et al. (Supplementary Figure 10).

2. In line 626, allele balance (AB) is used via a binomial test to classify variants as postzygotic (PZMs). While AB is only applied to a subset of variants, this still raises a multiple testing concern. If, as estimated, 470 variants required AB-based classification, a significance threshold of $\alpha = 0.05$ would imply approximately 23.5 false positives by chance. It **would be important to clarify how many variants remain classified as PZM after applying a multiple testing correction (e.g. FDR), or to justify why such correction is not necessary.**

This is a fair point. We applied the binomial test early in our variant calling pipeline to help determine downstream filtering thresholds (for example, we require more high-confidence reads of support for variants with AB significantly different from 0.5). When we make our final determination if a variant is germline or postzygotic in origin, we primarily base it on haplotype linkage in HiFi and ONT—if a variant is observed to be imperfectly linked to a parental haplotype, we call it postzygotic. However, in cases of conflicting HiFi and ONT assignment ($n=286$) and where read data cannot be phased ($n=184$), we do return to the binomial test to inform our final decision. As such, it would be more appropriate to apply multiple testing corrections to variants in our final callset.

We retested each variant, applied Benjamini-Hochberg correction, and passed the new germline/postzygotic determination into our pipeline. Of 6,958 retested variants, only six changed with multiple testing correction: three DNMs were reassigned as PZMs, and three PZMs were reassigned as DNMs.

Minor comments

- Figure 1A: it would be helpful to **clarify why a sibling was shown in the first example rather than a proband**, as this may confuse readers. A color legend should be added, as IGV color codes are not intuitive to all readers.

We picked a sibling to show as it was one of the clearest examples of a DNM near an informative phasing variant. To reduce confusion, we switched to a proband variant and added a color legend (see below).

C

- Reporting **gnomAD or TOPMed allele frequencies for DNMs** would clarify how rare these variants are. A gnomAD-based allele frequency threshold could also be incorporated as part of the filtering strategy, rather than relying solely on absence within the study cohort.

gnomADv4.1 allele frequencies are reported in Supplementary Table 6. We lifted 5,030 DNMs and 844 PZMs to GRCh38 and saw that 69.7% of DNMs and 73.5% of PZMs are not observed in gnomADv4.1. A further 23.8% of DNMs and 19.5% of PZMs have gnomAD AF < 1e-4, leaving a total of 328 DNMs and 59 PZMs more “common” variants in our dataset.

We opt to use absence in our study cohort as a filter for several reasons. First, for quads, the presence of a DNM call in a proband and sibling is a good indication of an inherited variant. Second, it provides a control for recurrent sequencing or alignment errors, as the same variants show up in the same data type across multiple samples. Finally, not all variants can be lifted to GRCh38, so these variants do not have a calculated gnomAD frequency.

gnomAD allele frequency for germline (left) and postzygotic (right) SNVs, after lifting over from T2T-CHM13v2.0 reference space to GRCh38 space. A total of 69.7% of DNMs and 73.5% of PZMs are not observed in gnomADv4.1.

We added the following text to the Methods (L704):

Although we evaluated the frequency of each variant in gnomAD4.1 (Supplementary Table 6), we did not use population frequency as a filter, as we wanted to specifically exclude sequencing and alignment artifacts arising from mapping long read data to T2TCHM13v2.0.

- Given that HiFi/Illumina data were generated from blood and ONT data from lymphoblastoid cell lines, **discordance in AB across platforms could be used to identify potential clonal expansion in blood.** This could be an informative additional analysis.

We evaluate AB concordance across platforms using a chi-squared test for each germline and postzygotic SNV (Supplementary Figure 3A,B). We find that roughly 89% of DNMs and 85% of PZMs are concordant across all three sequencing platforms.

We evaluated AB skewing in blood and cell-line data for 15 samples with at least 10X HiFi coverage derived from both sources. We examined 1,137 variants from these samples and restricted our analysis to reads with mapq>59 to remove any false mosaic events resulting from read multi-mapping. Based on Fisher's exact tests with Benjamini-Hochberg correction, we found only one variant with significantly different AB between blood- and cell-line-derived HiFi data (Supplementary Figure 3C,D). Although we did not observe a tight correlation between

blood and cell-line data, we also do not observe a systematic skew in either data type (Supplementary Figure 3E,F).

We added panels C-F to Supplementary Figure 3:

Supplementary Figure 3: Allele balance (AB) concordance scatterplots

A. AB of $n=5,145$ germline SNVs (DNMs) in PacBio HiFi, ONT, and Illumina read data. For each variant, a chi-squared test was used to determine whether its AB is concordant across platforms ($p>0.05$). A total of 89.1% of DNMs are concordant across platforms.

- B. AB of n=917 postzygotic SNVs (PZMs) in the same sequencing read data as A. AB concordance was again defined by chi-squared test ($p > 0.05$). A total of 85.7% of DNMs are concordant across platforms.
- C. AB in cell-line- and blood-derived HiFi read data (restricted to reads with mapq>59) for n=979 germline SNVs from 15 samples. Although we do not see the expected $x=y$ relationship between reads from both data sources, there is no evidence of systematic over- or underestimation between sources. Each variant was tested for concordant AB using a Fisher's exact test with Benjamini-Hochberg correction; none were significantly different ($p\text{-value} > 0.05$).
- D. AB in cell-line- and blood-derived HiFi read data (restricted to reads with mapq>59) for n=66 postzygotic SNVs from 15 samples. Similar to SNVs, we do not see a clear linear relationship between AB from data sources, but we also do not observe systematic bias. Each variant was tested for concordant AB using a Fisher's exact test with Benjamini-Hochberg correction; highlighted in yellow and circled in red is the single variant with significantly different AB across sources ($p=0.0193$).
- E. Bland-Altman plot showing the mean AB across data sources plotted against the difference between data sources for the same germline SNVs in (A). The mean difference in AB is 0.006, with a standard deviation of 0.188.
- F. Bland-Altman plot showing the mean AB across data sources plotted against the difference between data sources for the same postzygotic SNVs in (B). The mean difference in AB is 0.022, with a standard deviation of 0.197.

- A color legend is also missing from Figure 2B, which would improve interpretability.

We updated Figure 2B to include a color legend:

Authors' response to Reviewers:

Thank you very much for your careful suggestions; they have greatly helped us to improve the manuscript. We have responded to your final comments below.

1. I think the conclusion on line 125 is overstated and the Discussion is more appropriately circumspect. In particular, I am not sure how much evidence is provided by the lack of an age effect on PZMs, if the mutations occurred early in blood development, but I found the additional lines of evidence provided in the Discussion more convincing.

We modified the text to more precisely state our conclusion that we are likely not reporting somatic mutations that occurred later in life, but perhaps still include some from later in fetal development.

Original line 125:

Further, we do not see a significant correlation between sample age and PZM count, indicating that the impact of somatic mutations on our callset is minimal (Supplementary Figure 5).

New:

Further, we do not see a significant correlation between sample age and PZM count, indicating that the impact of postnatal somatic mutations on our callset is minimal (Supplementary Figure 5).

2. I wonder if the authors should consider a scenario in which the mutation is germline, but there is a gene conversion event in development (as mentioned line 411), and thus the mutation ends up on three haplotypes (especially given the finding that they see more PZM in SDs).

This is an interesting suggestion. We modified the text to further explore this possibility.

Original:

One possibility may be single-stranded DNA damage exists as heteroduplex in the sperm that cannot be adequately repaired until after conception. Even after fertilization, DNA transcription does not begin until the 4-8 cell stage, so some lesions may persist, constrained to the repair machinery available in the oocyte. Coupled with error-prone early cell divisions, the embryo may preferentially turn to repair mechanisms like allelic and interlocus gene conversion (IGC) to correct errors.

New:

One possibility may be single-stranded DNA damage exists as heteroduplex in the sperm that cannot be adequately repaired until after conception. Even after fertilization, DNA transcription does not begin until the 4-8 cell stage, so some lesions may persist, constrained to the repair machinery available in the oocyte. Coupled with error-prone early cell divisions, the embryo may preferentially turn to repair mechanisms like allelic and interlocus gene conversion (IGC) to correct errors. Another possibility is that some of these mutations may in fact be germline in origin, but a postzygotic gene conversion event rescues the original allele in a subset of daughter cells resulting in three haplotypes. We would expect to see such a pattern in high-identity regions such as SDs, and it may help to explain the observed paternal age effect.

3. In the paragraph starting on line 172, the authors may want to cite: <https://pubmed.ncbi.nlm.nih.gov/39806003/>

Thank you for this suggestion, it helped us to better contextualize our results.

Original:

Surprisingly, we notice a corresponding drop within the 10 bp of the 5' side of *de novo* CpG events suggesting a potential local epigenetic effect of DNM.

New:

Surprisingly, we notice a corresponding drop within the 10 bp of the 5' side of *de novo* CpG events suggesting a potential local epigenetic effect of DNM, as has been previously observed for somatic CpG mutations (Koch et al.).

4. The transmission rate reported for PZM on lines 384-384 seems oddly high, as I believe that the transmission rate should be equal to the allelic balance on average (since it is carried by that proportion of cells), so much less than 60% (closer to 20%).

The 60% statistic is not a per-PZM transmission rate, but the total number of PZMs we saw passed on to the next generation in the Platinum Genomes family. The samples in that pedigree each had six or eight children, so we saw a higher likelihood of passing a mutation on to the next generation than we would in a family with one or two children. In other words, of roughly 100 PZM calls in samples with sequenced offspring, about 60% were passed on to at least one offspring.

We saw the effect of low allele balance on the number of offspring that each PZM was passed on to – instead of seeing a PZM in about 50% of children (as you would expect for a germline DNM with allele balance of 0.5), we saw each PZM passed on to one or two children, reflecting the smaller proportion of cells that contain the variants.

This can be more clearly seen in Extended Data Figure 3 from Porubsky et al., excerpted below. In part a, each child inherits approximately 50% of their parent's germline SNVs, they only inherit 10-30% of their parent's postzygotic SNVs. Part b shows that PZMs with lower AB tend to be passed on to fewer children, although the data are quite noisy. Finally, part c shows that on average, each transmitted PZM is passed on to about 25% of offspring, matching AB expectations.

[editorial note: third party material redacted]

5. Starting line 389, the main text should discuss the lack of concordance between the mutational spectrum that they identify and others have reported, as they do in the reply to reviewers.

Original:

PZMs show unique mutational signatures and biases compared to germline mutations, likely

due to the different environment in which they arose. PZMs are depleted for CpG-associated mutations and are enriched for transversions when compared to those classified as germline (postzygotic Ti/Tv of 1.3 vs. germline Ti/Tv of 2.1). In addition, PZMs are enriched for A>C and A>T substitutions, as previously described.

New:

PZMs show unique mutational signatures and biases compared to germline mutations, likely due to the different environment in which they arose. PZMs are depleted for CpG-associated mutations and are enriched for transversions when compared to those classified as germline (postzygotic Ti/Tv of 1.3 vs. germline Ti/Tv of 2.1). Across published PZM callsets, there is not a consensus on the PZM spectrum, as Jonsson et al. report enrichments of A>C, C>A, and C>T mutations, while Sasani et al. report only an A>T enrichment, and Porubsky et al. report both an A>T enrichment and CpG>TpG depletion. In this study, we replicate the postzygotic enrichment of A>C and A>T mutations, as well as the depletion of CpG mutations.

Possible typos:

I think line 405 should read “embryonic repair of lesions that arose in the paternal germline” (i.e., lesions not mutations).

Thanks for noticing this; we made the fix.

Also line 431, what does “substitution mutations” refer to?

By substitution mutations, we mean single-nucleotide substitutions or SNVs. We changed the text to be more specific.

Original:

These findings suggest that SDs are particularly prone to substitution mutations early in embryogenesis.

New:

These findings suggest that SDs are particularly prone to single-nucleotide substitutions early in embryogenesis.